# RESISTING CONTEXTUAL INTERFERENCE IN RAG VIA PARAMETRIC-KNOWLEDGE REINFORCEMENT

**Chenyu Lin**[1,3]*      **Yilin Wen**[1]*      **Du Su**[2]      **Hexiang Tan**[2]
**Fei Sun**[2]      **Muhan Chen**[1]      **Chenfu Bao**[1,4][†]      **Zhonghou Lyu**[1][†]

[1]Baidu Inc.     [2]Institute of Computing Technology, Chinese Academy of Sciences
[3]Nankai University     [4]Tsinghua University
chenyulin@mail.nankai.edu.cn, {sudu,sunfei,tanhexiang21s}@ict.ac.cn
{wenyilin,chenmuhan01,baochenfu,lvzhonghou}@baidu.com
bcf25@mails.tsinghua.edu.cn

## ABSTRACT

Retrieval-augmented generation (RAG) improves performance on knowledge-intensive tasks but can be derailed by wrong, irrelevant, or conflicting retrieved text, causing models to rely on inaccurate evidence and cascade errors. We propose `Knowledgeable-R1`, a reinforcement-learning framework that explicitly trains large language models to use parametric knowledge (PK) to resist contextual interference while still exploiting external context when it is reliably helpful. `Knowledgeable-R1` introduces a joint sampling scheme that generates paired responses with and without retrieval, and learns both *local* advantages (within each decoding regime) and *global* advantages under the same input to quantify when to ignore misleading context versus adopt it. We employ an asymmetric advantage transformation that amplifies exploratory behaviors toward parametric knowledge. Experiments show that `Knowledgeable-R1` significantly improves robustness and reasoning accuracy in knowledge conflict scenarios and general RAG scenarios, outperforming SOTA baselines by +22.89% in counterfactual scenarios, and without degradation when the retrieved context is fully accurate. Datasets and code are available at https://github.com/lcy80366872/knowledgeable-R1.

## 1 INTRODUCTION

Retrieval-augmented generation (RAG) has become a prominent approach for enhancing large language models (LLMs) by integrating contextual knowledge, thereby mitigating hallucinations and reducing factual errors (Nakano et al., 2021; Gao et al., 2023). However, recent studies indicate that when contextual knowledge is introduced, LLMs can become overly reliant on this external information, suppressing their internal parametric knowledge. This phenomenon, known as *context dominance*, is particularly evident under conditions of noisy, counterfactual, or internally inconsistent evidence (Su et al., 2024; Xie et al., 2024b; Shi et al., 2023). Conflict-focused evaluations confirm that LLMs often adopt incorrect retrieved statements even when their parametric knowledge is correct and can lag behind retrieval-free reasoning when retrieval is imperfect (Bi et al., 2025a; Wang et al., 2025a; Wen et al., 2024). These findings highlight an imbalance in how models utilize knowledge.

A central challenge in Retrieval-Augmented Generation (RAG) systems is determining when to rely on contextual knowledge (CK) and when to revert to parametric knowledge (PK), as well as how to integrate the two in a stable and faithful manner. 1) Prompting approaches help guide the model to validate or filter the context while combining it with parametric knowledge, which improves the coherence of the output (Ding et al., 2024; Cheng et al., 2024; Press et al., 2023; Wang et al., 2023; He et al., 2024; Wang et al., 2025a). 2) Decoding-based approaches, such as those proposed by (Bi et al., 2025b), adjust the token distribution during generation to mitigate conflicts between external context and parametric knowledge. While effective in some scenarios, both prompting and decoding methods add computational complexity and lack a generalizable decision rule for managing diverse contextual situations. 3) Fine-tuning methods, such as Self-RAG (Asai et al., 2024) and InFO-RAG (Xu et al., 2024), train LLMs to implicitly learn decision rules for knowledge utilization. However, these methods often require complex data annotation pipelines, which can limit flexibility and scalability.

---

*Equal contribution.     [†]Corresponding author.

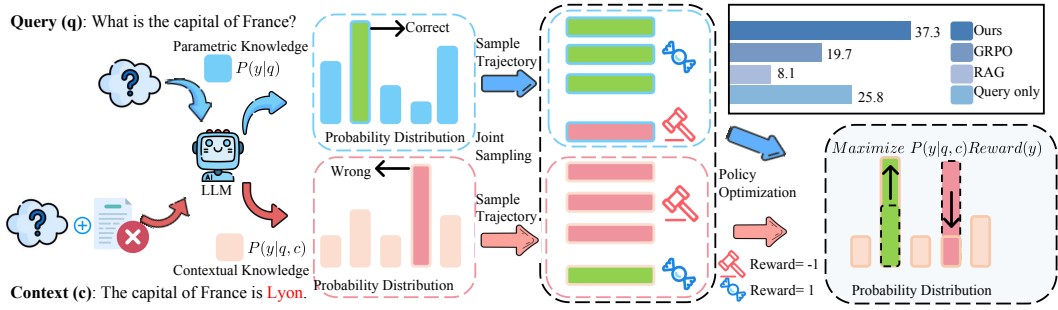

Figure 1: Overview of `Knowledgeable-R1`, which integrates parametric (blue) and contextual (pink) sampling with policy optimization to ensure robust reasoning under misleading context. Green and red blocks indicate correct and incorrect responses, respectively. The inset chart displays accuracy on the ConFiQA-MC adversarial scenario (S2), where `Knowledgeable-R1` (**37.3%**) significantly outperforms GRPO w/ RAG (19.7%), query-only prompting (25.8%), and RAG prompting (8.1%); see Table 2.

When contextual knowledge appears reliable but is actually incorrect, LLMs should ignore it and decide whether to fall back on parametric knowledge. This behavior is rare but crucial, as current LLMs tend to rely more on context when faced with conflicting or misleading information. Reinforcement learning (RL) can assist by adjusting the probability rankings of outputs, encouraging the LLM to explore these rare but critical decisions, such as when to fall back on parametric knowledge (Stiennon et al., 2020; Ramamurthy et al., 2023). Therefore, we believe RL has the potential to help LLMs learn how to effectively balance and utilize both parametric and contextual knowledge. Recently, models like OpenAI-o1 (Jaech et al., 2024) and DeepSeek-R1 (Guo et al., 2025b) have employed RL techniques such as PPO (Schulman et al., 2017) and GRPO (Shao et al., 2024) to enhance LLMs' logical reasoning and problem-solving abilities through experience and feedback. However, in RAG systems, these methods are limited by their sampling space and objectives, which leads to a primary focus on context-aware reasoning while overlooking the importance of incorporating parametric knowledge.

The goal of this work is to develop a reinforcement learning framework that explores both contextual and parametric knowledge after retrieval input. We call our method `Knowledgeable-R1` because it is a knowledge-aware reasoning approach that uses parametric knowledge to mitigate contextual interference in the RAG system. A conceptual overview of our framework is presented in Figure 1. Specifically, `Knowledgeable-R1` enables LLMs to (1) explore both parametric and contextual knowledge through joint sampling, (2) distinguish the relative merits of these two knowledge types using locally and globally defined advantages, and (3) mitigate penalties for less likely parametric outputs by applying an adaptive asymmetric advantages transformation. We conducted experiments across five scenarios, demonstrating that `Knowledgeable-R1` outperforms a range of RAG approaches by a significant margin. By effectively utilizing both parametric and contextual knowledge, `Knowledgeable-R1` shows significant improvements in adversarial context settings, achieving **+22.9%** improvement over GRPO baselines in counterfactual scenarios (Table 3) and a **30.47%** improvement over original RAG prompting, while maintaining strong performance in reliable contexts.

## 2 RELATED WORK

### 2.1 LIMITATIONS OF EXISTING RAG ROBUSTNESS METHODS

Recent studies establish that LLMs can store substantial parametric knowledge (PK), but their ability to use external contextual knowledge (CK) degrades under noisy inputs. Many studies have analyzed this phenomenon and found that it may be caused by position biases (Liu et al., 2024; Hu et al., 2025) or interference when PK and CK conflict (Farahani et al., 2024). These observations highlight the need for methods that can dynamically retain reliable PK when CK is misleading, while still leveraging CK when beneficial.

A significant thread in RAG research focuses on robustness to irrelevant or adversarial passages. Methods include noise-injected training (Yoran et al., 2024), attention masking (Fang et al., 2024),

and adversarial filtering (Shi et al., 2024; Cohen-Wang et al., 2024). Evaluation benchmarks now systematically stress *conflicting evidence* scenarios (Wang et al., 2025b; Tan et al., 2024). However, these approaches largely operate at the *passage level* through pre-retrieval filtering or post-retrieval weighting, lacking a mechanism to actively suppress harmful context during generation. This limitation is particularly acute when misleading information appears plausible, as models tend to over-prioritize context even when it contradicts their parametric knowledge (Farahani et al., 2024; Gao et al., 2023).

## 2.2 REINFORCEMENT LEARNING FOR DECISION POLICIES IN LLM REASONING

Beyond supervised fine-tuning, reinforcement learning (RL) has emerged as a powerful paradigm for shaping model behavior without extensive human annotation. Recent work demonstrates RL's effectiveness in guiding reasoning processes, such as generating internal rationales (Zelikman et al., 2024; Li et al., 2025b) or stabilizing long chain-of-thought reasoning (Yu et al., 2025; Chen et al., 2025; Cheng et al., 2025; Jin et al., 2025; Li et al., 2025a; Song et al., 2025). However, these methods primarily focus on *reasoning structure* rather than on *parametric and contextual knowledge-aware learning for LLMs themselves*.

## 3 METHOD

Our objective is to enable a single language model to appropriately leverage parametric knowledge (PK) versus contextual knowledge (CK) in retrieval-augmented generation (RAG). The key challenge lies in the varying reliability of retrieved context, which can be helpful, redundant, or misleading. An ideal LLM should:

- Select the most accurate answer from its parametric knowledge when no context is provided (*parametric-only correctness*).
- Select the most accurate answer from its contextual knowledge when context is provided (*context-aware correctness*).
- Fall back on its correct parametric knowledge when faced with conflicting or noisy context, ensuring robustness to misleading information (*robustness to misleading context*).

To achieve this, we propose a multi-objective reinforcement learning framework that trains LLMs to simultaneously optimize these three goals.

### 3.1 TASK DEFINITION

We formulate the problem as a token-level reinforcement learning task. Let $\mathcal{V}$ be the vocabulary, and let a prompt $x$ be the input sequence. The prompt can be of two types: $p$, containing only the query $q$, or $p'$, containing both the query $q$ and a retrieved context passage $c$. At each decoding step $t$, the model (parameterized by $\theta$) produces a probability distribution over $\mathcal{V}$ based on the prompt and previously generated tokens $o_{<t}$:

$$\pi_\theta(o_t|x, o_{<t}) = \text{softmax}(f_\theta(x, o_{<t}))$$

The goal is to optimize $\theta$ such that the generated sequence $o = (o_1, \ldots, o_T)$ maximizes the expected reward, which reflects answer correctness and appropriate knowledge utilization.

### 3.2 THREE SAMPLING/DECODING POLICIES.

Let $p$ denote the *query-only* prompt and $p'$ denote the *query+context* prompt. As we illustrate in Section 1, GRPO uses a single sampling policy during LLM training, making it difficult for the LLM to explore parametric-aware answers in the combined retrieval input $p'$. In contrast, our method defines three distinct policies for each query $q$ over the next-token distributions:

$$\pi_\theta(o_t \mid p, o_{<t}), \quad \pi'_\theta(o'_t \mid p', o'_{<t}), \quad \hat{\pi}_\theta(o_t \mid p', o_{<t}).$$

**Notation.** We use $o$ to denote a token sequence generated when the *current* policy is conditioned on $p$ (query-only input), and $o'$ to denote a token sequence generated when the *current* policy is

conditioned on $p'$ (query+context input). We write $o_{<t}$ for the prefix up to step $t-1$, and $o_t$ for the token at step $t$. With these, the three policy types have the following inputs/outputs and intended behaviors:

- **PK (Parametric)**: input $p =$ query, output $o$ (answer from parametric knowledge).
- **CK (Context-aware)**: input $p' =$ query+context, output $o'$ (answer using context).
- **RPK (Robust-PK)**: input $p' =$ query+context, output $o$ (answer consistent with PK).

CK and RPK share the same input $p'$ but target different behaviors: CK exploits reliable context; RPK *stays on the PK trajectory* when context is noisy/conflicting. PK and RPK share the same output type $o$, but are trained under different conditioning (without vs. with context). At inference, the model does not explicitly switch controllers; the learned token distributions encode when to follow context or implicitly revert to PK.

**Clarification on RPK sampling.** RPK does not generate an independent answer. For each query, we first sample a PK trajectory $o^{\mathrm{pk}} = (o_1^{\mathrm{pk}}, \ldots, o_T^{\mathrm{pk}})$ from $\pi_\theta(\cdot \mid p, \cdot)$ (query-only). In the RPK branch, the same token sequence $o^{\mathrm{pk}}$ is used as the *target*, but evaluated under the query+context prompt $p'$: at each step $t$, we compute $\pi_\theta(o_t^{\mathrm{pk}} \mid p', o_{<t}^{\mathrm{pk}})$ and maximize its log-probability under the RPK reward. In other words, RPK *re-evaluates* the PK answer under $p'$ and encourages the model to maintain PK tokens even when misleading context is present.

Table 1: I/O and behavior of the three decoding policies.

| Policy type | Input prompt | Output type | Intended behavior |
|---|---|---|---|
| PK | $p =$ query | $o \in$ answer from parametric knowledge | Parametric-only answer |
| CK | $p' =$ query+context | $o' \in$ answer using context | Use context when helpful |
| RPK | $p' =$ query+context | $o \in$ answer consistent with PK | Fallback PK answer under misleading context |

### 3.3 Advantage Calculation with Local and Global Normalization

To effectively balance the three objectives, we introduce a novel advantage calculation scheme that combines within-objective and cross-objective comparisons. The components are:

- **Local Advantage** (within-policy, within-input): compares trajectories that share the same input *and* the same output policy type.
- **Global Advantage** (same input state, cross-output): compares CK and RPK trajectories under $p'$ in a unified pool, deciding whether to *use* context or *fall back* to PK-style decoding.

The global advantage mechanism enables effective differentiation between group trajectories, ensuring that rewards can still capture *cross-source* preferences (CK vs. RPK) when all trajectories within a group are uniformly good or bad. This maintains meaningful feedback for different knowledge types.

**Notation.** Rewards $R(\cdot)$ are computed per trajectory based on answer correctness (e.g., EM). Let rewards be $R_i^{\mathrm{pk}}$ for PK trajectories $o_i \sim \pi_\theta(\cdot \mid p, \cdot)$, $R_j^{\mathrm{ck}}$ for CK trajectories $o_j' \sim \pi_\theta'(\cdot \mid p', \cdot)$, and $R_i^{\mathrm{rpk}}$ for RPK trajectories $\tilde{o}_i \sim \hat{\pi}_\theta(\cdot \mid p', \cdot)$. Define the *global (same-input)* pool under $p'$ as $\mathcal{U}_{p'} = \{R_j^{\mathrm{ck}}\} \cup \{R_i^{\mathrm{rpk}}\}$.

To address the "overcorrection" problem, when CK is reliable but slightly differs from PK (e.g., when factual details like a new capital city are updated), the LLM should trust CK more due to its timeliness. Thus, we design distinct advantages for PK, CK, and RPK.

**PK (Parametric).** The advantage of PK is defined as $A_i = A_i^{\mathrm{pk\text{-}local}}$, since PK relies on query-only input:

$$A_i^{\mathrm{pk\text{-}local}} = \frac{R_i^{\mathrm{pk}} - \mathrm{mean}(\{R_i^{\mathrm{pk}}\})}{\mathrm{std}(\{R_i^{\mathrm{pk}}\}) + \varepsilon}.$$

This term encourages the query-only parametric answer to be as accurate as possible.

**CK (Context-aware).** The advantage of CK is $A_j' = A_j^{\mathrm{ck\text{-}local}} + A_j^{\mathrm{ck\text{-}global}}$, combining both *local* and *global* terms:

$$A_j^{\text{ck-local}} = \frac{R_j^{\text{ck}} - \text{mean}(\{R_j^{\text{ck}}\})}{\text{std}(\{R_j^{\text{ck}}\}) + \varepsilon}, \quad A_j^{\text{ck-global}} = \frac{R_j^{\text{ck}} - \text{mean}(\mathcal{U}_{p'})}{\text{std}(\mathcal{U}_{p'}) + \varepsilon}.$$

**RPK (Robust-PK under Context).** The advantage of RPK is defined as $\hat{A}_i = \hat{A}_i^{\text{global}}$. RPK focuses solely on the *global* comparison under the same input $p'$:

$$\hat{A}_i^{\text{global}} = \frac{R_i^{\text{rpk}} - \text{mean}(\mathcal{U}_{p'})}{\text{std}(\mathcal{U}_{p'}) + \varepsilon}.$$

**Summary.** PK has a local advantage, ensuring accuracy based on the query input. CK combines both local and global advantages, prioritizing context when both knowledge types are correct, as context is frequently updated and more reliable. RPK has only a global advantage, comparing PK's performance in the same input $p'$ to maintain it as a fallback when context is misleading. This balance ensures the model uses context effectively without over-relying on it when PK is more reliable.

## 3.4 KNOWLEDGE BALANCE MODULATION

During training, context-aware (CK) paths often get higher rewards than robust parametric knowledge (RPK) paths due to the usefulness of retrieved contexts, creating a bias towards context. This can make the model rely too much on CK and weaken its ability to use parametric knowledge in noisy or conflicting situations. To address this, we introduce an asymmetric advantage transformation for RPK paths, reducing the penalty when PK-based decoding is slightly worse than context-following, ensuring parametric knowledge remains a viable fallback.

We define a modulation function $T$ that transforms the RPK advantages $\hat{A}_i$:

$$T(\hat{A}_i; \beta) = \begin{cases} \hat{A}_i, & \text{if } \hat{A}_i > 0, \\ \beta \cdot \hat{A}_i, & \text{if } \hat{A}_i \leq 0, \end{cases}$$

where $\beta \in [0.01, 1]$ is a modulation coefficient. When $\beta < 1$, negative advantages (indicating poor performance of PK decoding) are reduced, making the model less sensitive to occasional mistakes when relying on parametric knowledge.

To dynamically balance the exploration of parametric and contextual knowledge, we adapt $\beta$ during training based on the relative performance of CK and RPK trajectories. Let $\mathcal{B}$ denote the current mini-batch of training examples. We compute the total advantage for each knowledge type:

$$S_{\text{ck}} = \sum_{j \in \mathcal{B}_{\text{ck}}} A_j', \quad S_{\text{rpk}+} = \sum_{i \in \mathcal{B}_{\text{rpk}+}} \hat{A}_i, \quad S_{\text{rpk}-} = \sum_{i \in \mathcal{B}_{\text{rpk}-}} \hat{A}_i,$$

where $\mathcal{B}_{\text{ck}}$ is the set of CK trajectories, $\mathcal{B}_{\text{rpk}+}$ is the subset of RPK trajectories with positive advantages, and $\mathcal{B}_{\text{rpk}-}$ is the subset with non-positive advantages.

We then update $\beta$ to maintain balance between the two knowledge sources:

$$\beta \leftarrow \text{clip}\left(\frac{S_{\text{ck}} - S_{\text{rpk}+}}{S_{\text{rpk}-}}, 0.01, 1\right),$$

The update rule adjusts the penalty coefficient $\beta$ based on the performance gap between context-aware (CK) and robust parametric knowledge (RPK). When CK outperforms RPK, $\beta$ decreases, reducing penalties for negative RPK advantages and encouraging more exploration of RPK. As the gap narrows, $\beta$ increases, making RPK training more cautious.

When context is highly beneficial (i.e., $S_{\text{rpk}-}$ is much larger than $S_{\text{rpk}+}$), reducing penalties for poor RPK performance ($\beta$ decreases) allows the model to keep RPK as a fallback option. This method, similar to *reward shaping* (Ng et al., 1999; Devlin et al., 2011), ensures that parametric knowledge stays usable. It also resembles focused exploration techniques (Schulman et al., 2015; Espeholt et al., 2018) that prevent forgetting learned behaviors. While this introduces some bias in the gradient estimates, it helps balance the model's tendency to over-rely on helpful context from the training data, allowing it to resist misleading context when needed.

## 3.5 POLICY OPTIMIZATION

We adopt PPO-style updates with clipping. For each trajectory, we compute the probability ratio between the current and old policies:

$$r_{i,t}^{\mathrm{PK}} = \frac{\pi_\theta(o_{i,t} \mid p,\ o_{i,<t})}{\pi_{\theta_{\mathrm{old}}}(o_{i,t} \mid p,\ o_{i,<t})}, \quad r_{j,t}^{\mathrm{CK}} = \frac{\pi'_\theta(o'_{j,t} \mid p',\ o'_{j,<t})}{\pi'_{\theta_{\mathrm{old}}}(o'_{j,t} \mid p',\ o'_{j,<t})}, \quad r_{i,t}^{\mathrm{RPK}} = \frac{\hat\pi_\theta(o_{i,t} \mid p',\ o_{i,<t})}{\hat\pi_{\theta_{\mathrm{old}}}(o_{i,t} \mid p',\ o_{i,<t})}.$$

where $(x, o_t)$ are the corresponding prompts and tokens.

The objective for each component is defined as follows:

$$J_{\mathrm{PK}} = \frac{1}{n_{\mathrm{pk}}} \sum_{i=1}^{n_{\mathrm{pk}}} \sum_{t=1}^{|o_i|} \min \left[ r_{i,t}^{\mathrm{PK}} A_i,\ \mathrm{clip}(r_{i,t}^{\mathrm{PK}}, 1-\epsilon, 1+\epsilon)\, A_i \right],$$

$$J_{\mathrm{CK}} = \frac{1}{n_{\mathrm{ck}}} \sum_{j=1}^{n_{\mathrm{ck}}} \sum_{t=1}^{|o'_j|} \min \left[ r_{j,t}^{\mathrm{CK}} A'_j,\ \mathrm{clip}(r_{j,t}^{\mathrm{CK}}, 1-\epsilon, 1+\epsilon)\, A'_j \right], \tag{1}$$

$$J_{\mathrm{RPK}} = \frac{1}{n_{\mathrm{pk}}} \sum_{i=1}^{n_{\mathrm{pk}}} \sum_{t=1}^{|o_i|} \min \left[ r_{i,t}^{\mathrm{RPK}} T(\hat A_i),\ \mathrm{clip}(r_{i,t}^{\mathrm{RPK}}, 1-\epsilon, 1+\epsilon)\, T(\hat A_i) \right].$$

The total objective is the weighted sum of the individual objectives:

$$\mathcal{J}(\theta) = \lambda_{\mathrm{pk}} J_{\mathrm{PK}} + \lambda_{\mathrm{ck}} J_{\mathrm{CK}} + \lambda_{\mathrm{rpk}} J_{\mathrm{RPK}} \tag{2}$$

with $\lambda_{\mathrm{pk}} = \lambda_{\mathrm{ck}} = \lambda_{\mathrm{rpk}} = 1.0$ in our experiments. We use $\epsilon = 0.2$ for clipping.

The goal of our method is to optimize $\theta$ to maximizes $\mathcal{J}(\theta)$, the full training procedure is outlined in Algorithm 1 in Appendix L.

## 4 EXPERIMENTS

We evaluate whether the model can *leverage parametric knowledge* to produce accurate answers despite *external contextual interference*, and remain robust when context knowledge is correct.

### 4.1 EXPERIMENTAL SETUP

We test robustness under five contextual conditions with increasing difficulty: **Scenario I (S1)**: Correct contextual knowledge (Section 4.2.1); **Scenario II (S2)**: Adversarial contextual knowledge (Section 4.2.1); **Scenario III (S3)**: Self-conflicting contextual knowledge (Section 4.2.1); **Scenario IV (S4)**: Irrelevant contextual knowledge (Section 4.2.1); **Scenario V (S5)**: Partially relevant contextual knowledge (Section 4.2.1). These scenarios reflect real-world retrieval situations where irrelevant or conflicting context can mislead generation.

#### 4.1.1 REASONING INSTRUCTION AND BASELINES

We evaluate with `Qwen2.5-3B-Instruct`, `Qwen2.5-7B-Instruct`, `Qwen2.5-14B-Instruct` and `Llama3.1-8B-Instruct`. Baselines include query-only prompting, RAG prompting, Astute-RAG (a prompting method for imperfect retrieval (Wang et al., 2025a)), CK-PLUG (Bi et al., 2025c), SFT, and GRPO (Guo et al., 2025a) with RAG. We evaluate using exact match (EM) (Jin et al., 2025). To ensure fairness, all methods use the same total training data and are trained for one epoch. For GRPO and `Knowledgeable-R1`, we use a global batch size of 128, a rollout batch size of 32, a rollout temperature of 1, and a learning rate of $1 \times 10^{-6}$. All experiments run on 8 H100 GPUs, using the same system prompt for both training and inference, where the inference input combines the query and context. For further experiment details, see Appendix I, J, and K.

**Baseline clarifications. CK-PLUG** (Bi et al., 2025c) is a plug-and-play decoding method that detects knowledge conflicts via confidence gain and adjusts token probabilities for conflict spans, enabling

fine-grained control over parametric vs. contextual reliance. **GRPO w/ RAG** trains only the CK objective with query+context rollouts (no PK/RPK branches), using the same number of total rollouts as `Knowledgeable-R1`. **SFT** is a strong fine-tuning baseline trained on the same mixture of good/bad context data as `Knowledgeable-R1`, with explicit conflict-handling instructions. Our method splits rollouts into PK, CK, and RPK groups; roughly half the rollout budget is allocated to query-only PK trajectories, making the two methods directly comparable.

### 4.1.2 DATASETS

We introduce **KnowQA** to test methods across the five scenarios. First, we test parametric knowledge using query-only prompts (Appendix H), where a correct answer means the model has the relevant PK. Then, we create contextual inputs using the top-5 retrieved paragraphs. This method is suitable for knowledge-intensive tasks. KnowQA includes scientific, factual, and commonsense questions, organized by scenario:

- **Correct and Wrong Context**. We adapt **PC** (PC-QA, PC-MR, PC-MC) and **NC** (NC-QA, NC-MR, NC-MC) from **ConFiQA** (Xie et al., 2024a) for supporting vs. opposing evidence. QA stands for single-hop QA, MR for multi-hop reasoning, and MC for multiple conflicting inputs. We check the correctness of the context by looking for made-up sources. ConFiQA is a counterfactual-retrieval benchmark with three sub-datasets (QA/MR/MC).
- **Conflict Context**. We create **SC** from ConFiQA using conflicting evidence and test whether models can handle contradictions by showing both positive and negative examples together.
- **Irrelevant Context**. We use **ExplainPE** (Wen et al., 2024), a medical QA dataset from the Chinese National Licensed Pharmacist Examination. It *tests the impact of knowledge mismatches on model performance*, where adding retrieved documents can *lower* accuracy compared to query-only.
- **Partly Irrelevant Context**. We use **HotpotQA** (Yang et al., 2018), **2WikiMultiHopQA** (Ho et al., 2020), and **MuSiQue** (Trivedi et al., 2022), which include valid evidence mixed with distractors.

### 4.1.3 RETRIEVAL SETUP

We use dataset-provided contexts without any additional retriever. The contexts (including both correct and incorrect passages) are directly taken from the benchmark construction, without re-indexing or re-retrieval from a larger corpus. We adopt the top-5 paragraphs provided by these benchmarks, preserve their original order, and only truncate when the backbone model's maximum length is exceeded. For ConFiQA training, correct and incorrect contexts are mixed in equal proportion.

### 4.2 EXPERIMENTAL RESULTS

We evaluate `Knowledgeable-R1` in various contexts and find that it effectively selects reliable evidence while ignoring misleading or irrelevant information, especially when parametric knowledge (PK) is sufficient. This leads to significant improvements in challenging interference settings (S2-S5) without harming performance in correct contexts (S1). Consistent results are observed across models of different sizes (3B, 7B, 8B, 14B), with detailed results for `Qwen2.5-7B-Instruct` and `Llama3.1-8B-Instruct` in Tables 2 and 3, and similar patterns for other models (see Appendix M). Our experimental analysis highlights three key strengths of `Knowledgeable-R1`:

- **Robustness to Incorrect Context.** Significant improvements in adversarial (S2), conflicting (S3), and irrelevant (S4) settings demonstrate the method's ability to handle unreliable context.
- **Preservation of Correct Context Benefits.** No performance drop occurs when the context is accurate (S1), keeping it competitive with context-specific baselines.
- **Effective Context Selection.** Large gains on PK-answerable questions, particularly in mixed-context scenarios (S5), highlight the method's ability to filter relevant from irrelevant information.

### 4.2.1 PERFORMANCE ACROSS CONTEXTUAL SCENARIOS

**Scenario I: Correct Contextual Knowledge (S1).** When given accurate evidence, `Knowledgeable-R1` keeps the performance improvements from correct context. As shown in Table 2 (left columns), for `Qwen2.5-7B-Instruct`, Knowledgeable-R1 achieves **80.90%** on PC-QA and **75.51%** on PC-MC, staying competitive with the best baseline (GRPO w/ RAG). Similarly, for `Llama3.1-8B-Instruct`, it reaches **80.03%** on PC-QA and **80.24%** on PC-MC.

Table 2: Overall accuracy across five contextual scenarios: correct (S1), wrong (S2), conflict (S3), irrelevant (S4), and partly-irrelevant (S5). Best results are in **bold**, second best are underlined. The "improve" row shows gains over RAG prompting.

| Method | Correct (S1) | | | Wrong (S2) | | | Conflict (S3) | Irrelevant (S4) | Partly Irrelevant (S5) | | |
| --- | --- | --- | --- | --- | --- | --- | --- | --- | --- | --- | --- |
| | PC-MR | PC-MC | PC-QA | NC-MR | NC-MC | NC-QA | SC | ExplainPE | HotPotQA | 2Wiki MultiHopQA | Musique |
| `Qwen2.5-7B-Instruct` | | | | | | | | | | | |
| Query-only prompting | 27.72% | 24.66% | 31.67% | 25.93% | 25.82% | 32.31% | 29.67% | 64.45% | 20.90% | 25.54% | 4.36% |
| RAG prompting | 65.68% | 66.39% | 74.35% | 13.47% | 8.06% | 11.31% | 59.50% | 62.21% | 20.36% | 22.53% | 6.41% |
| CK-PLUG (Bi et al., 2025b) | 64.69% | 66.55% | 78.66% | 11.62% | 8.06% | 7.92% | 55.00% | 55.00% | 22.74% | 24.76% | 6.25% |
| Astute (Wang et al., 2025a) | 65.51% | 66.05% | 77.62% | 12.79% | 7.07% | 10.34% | 54.20% | 56.74% | 17.87% | 20.35% | 6.29% |
| SFT | 71.95% | 77.70% | 74.70% | 24.92% | 21.05% | 21.97% | 68.50% | 66.60% | 30.14% | 32.20% | 11.75% |
| GRPO w/ RAG | 77.56% | 77.36% | 80.03% | 26.94% | 19.74% | 26.01% | 75.33% | 66.50% | 27.93% | 33.95% | 11.79% |
| `Knowledgeable-R1` | 75.08% | 75.51% | 80.90% | 43.94% | 37.34% | 29.40% | 76.33% | 67.57% | 31.45% | 37.52% | 12.04% |
| improve | +9.41% | +9.12% | +6.54% | +30.47% | +29.28% | +18.09% | +15.92% | +5.36% | +11.09% | +14.99% | +5.63% |
| `Llama3.1-8B-Instruct` | | | | | | | | | | | |
| Query-only prompting | 29.37% | 26.18% | 39.93% | 27.10% | 27.63% | 42.65% | 32.08% | 43.26% | 20.69% | 21.02% | 6.16% |
| RAG prompting | 64.85% | 61.99% | 76.42% | 22.90% | 16.28% | 24.88% | 61.17% | 39.16% | 24.44% | 23.50% | 8.19% |
| CK-PLUG (Bi et al., 2025b) | 54.79% | 58.45% | 69.71% | 12.12% | 9.05% | 17.29% | 42.00% | 31.54% | 22.35% | 24.63% | 5.25% |
| Astute (Wang et al., 2025a) | 65.84% | 64.86% | 77.97% | 17% | 9.05% | 17.29% | 59.83% | 40.14% | 1.65% | 30.26% | 9.64% |
| SFT | 72.88% | 79.22% | 73.84% | 42.59% | 35.53% | 35.86% | 70.12% | 47.17% | 33.59% | 38.24% | 13.36% |
| GRPO w/ RAG | 78.05% | 79.73% | 82.62% | 41.58% | 35.69% | 39.26% | 76.58% | 47.56% | 34.84% | 41.22% | 16.59% |
| `Knowledgeable-R1` | 73.76% | 80.24% | 80.03% | 55.39% | 41.12% | 44.59% | 73.67% | 49.61% | 37.06% | 45.37% | 14.69% |
| improve | +8.91% | +18.25% | +3.61% | +32.49% | +24.84% | +19.71% | +15.41% | +10.45% | +12.62% | +21.87% | +6.50% |

These results show that learning to manage noisy contexts doesn't reduce the use of reliable evidence, addressing a key concern in robust RAG systems.

**Scenario II: Adversarial Contextual Knowledge (S2).** This scenario highlights the weaknesses of naive RAG and the benefits of our method. When the context is adversarial, RAG prompting causes a huge drop in performance—down to **13.47%/8.06%/11.31%** (NC-MR/MC/QA) on Qwen2.5-7B-Instruct, much lower than the query-only baseline. Knowledgeable-R1 greatly improves this, increasing performance by **+30.47/+29.28/+18.09** percentage points and outperforming GRPO w/ RAG (e.g., **43.94%** vs. **26.94%** on NC-MR). The consistent gains of **+32.49/+24.84/+19.71** on Llama3.1-8B-Instruct suggest that the method effectively learns to ignore misleading evidence and rely on parametric knowledge when needed.

**Scenario III: Self-Conflicting Contextual Knowledge (S3).** When the context has internal contradictions, Knowledgeable-R1 performs competitively: **76.33%** compared to **75.33%** for GRPO w/ RAG on Qwen2.5-7B-Instruct, and **73.67%** compared to **76.58%** on llama3.1-8B. Though the improvements are small, the consistent gains across models show stable handling of conflicting information. Resolving contradictions within the context seems difficult for all methods, indicating an area for future improvement.

**Scenario IV: Irrelevant Contextual Knowledge (S4).** In the **ExplainPE** benchmark, where the context is completely unrelated, we see the typical "RAG-hurt" effect: query-only (**64.45%**) performs better than RAG prompting (**62.21%**) on Qwen2.5-7B-Instruct. Knowledgeable-R1 achieves **67.57%**, actively rejecting irrelevant context and outperforming both methods. This shows that the method can recognize when context isn't helpful for the query.

**Scenario V: Partially Relevant Contextual Knowledge (S5).** When valid evidence is mixed with distractors, Knowledgeable-R1 performs strongest or nearly strongest. On Qwen2.5-7B-Instruct, it achieves **31.45%** on HotpotQA, **37.52%** on 2WikiMultiHopQA, and **12.04%** on MuSiQue, with significant gains over RAG prompting. **Notably, the strong performance on 2WikiMultiHopQA and MuSiQue is achieved without fine-tuning on these datasets**, showing that the method can generalize to new evidence types and distractor patterns beyond the training data (HotpotQA). This ability to generalize across datasets is especially useful for real-world applications, where the evidence may differ from the training data.

### 4.2.2 ANALYSIS ON PARAMETRIC-KNOWLEDGE ANSWERABLE SUBSET

We define the *parametric-knowledge answerable subset* as questions where the model provides correct answers using query-only prompting, showing that it has enough parametric knowledge. When focusing on this subset, all methods improve, but Knowledgeable-R1 shows the biggest gains when the context is wrong or mixed (Table 3). Specifically, on NC-MR, NC-MC, and NC-QA, our method achieves an average improvement of 22.89% over GRPO w/ RAG.

Table 3: Accuracy on the parametric-knowledge answerable subset across all scenarios. Best results are in **bold**, second best are underlined.

| | Correct (S1) | | | Wrong (S2) | | | Conflict (S3) | Irrelevant (S4) | Partly Irrelevant (S5) | | |
| --- | --- | --- | --- | --- | --- | --- | --- | --- | --- | --- | --- |
| Method | PC-MR | PC-MC | PC-QA | NC-MR | NC-MC | NC-QA | SC | ExplainPE | HotPotQA | 2Wiki MultiHopQA | Musique |
| `Qwen2.5-7B-Instruct` | | | | | | | | | | | |
| RAG prompting | 85.71% | 86.99% | 94.02% | 32.47% | 19.75% | 28.50% | 87.08% | 83.64% | 48.84% | 41.16% | 20.75% |
| CK-PLUG (Bi et al., 2025b) | 79.76% | 83.56% | 95.65% | 29.87% | 18.47% | 18.50% | 82.87% | 77.58% | 55.30% | 42.78% | 20.75% |
| Astute (Wang et al., 2025a) | 86.31% | 87.67% | 92.39% | 31.17% | 19.11% | 24.50% | 81.18% | 73.03% | 41.73% | 34.50% | 20.75% |
| GRPO w/ RAG | 93.21% | 93.15% | 97.28% | 57.61% | 46.50% | 62.50% | 93.54% | **84.39%** | 66.54% | 55.42% | 45.28% |
| `Knowledgeable-R1` | **95.83%** | **95.21%** | **97.83%** | **89.61%** | **79.62%** | **66.00%** | **96.07%** | 83.18% | **78.49%** | **66.72%** | **66.98%** |
| improve | +10.12% | +8.22% | +3.81% | +57.14% | +59.87% | +37.50% | +8.99% | -0.46% | +29.65% | +25.56% | +46.23% |
| `Llama3.1-8B-Instruct` | | | | | | | | | | | |
| RAG prompting | 89.89% | 87.74% | 96.98% | 52.17% | 37.50% | 48.48% | 87.01% | 67.95% | 55.16% | 39.92% | 34.90% |
| CK-PLUG(Bi et al., 2025b) | 80.90% | 82.58% | 90.09% | 30.43% | 21.43% | 32.58% | 67.01% | 44.24% | 49.15% | 42.79% | 16.11% |
| Astute (Wang et al., 2025a) | 91.01% | 89.03% | 97.41% | 35.40% | 20.24% | 33.33% | 83.12% | 70.20% | 3.12% | 52.25% | 35.57% |
| GRPO w/ RAG | **96.07%** | **94.19%** | **98.71%** | 72.67% | 71.43% | 70.83% | 94.81% | 74.94% | 73.17% | 68.94% | 61.74% |
| `Knowledgeable-R1` | 94.38% | 93.55% | **98.71%** | **88.82%** | **75.60%** | **81.82%** | **96.62%** | **78.10%** | **83.55%** | **81.69%** | **71.14%** |
| improve | +4.49% | +5.81% | +1.73% | +36.65% | +38.10% | +33.34% | +9.61% | +10.15% | +28.39% | +41.77% | +36.24% |

Table 4: Ablation study of `Knowledgeable-R1` components. TITE: Both parametric and contextual knowledge correct; TIFE: parametric correct, contextual wrong; FITE: parametric wrong, contextual correct; FIFE: Both wrong. Best results are in **bold**, largest performance drops are underlined.

| | PC-MR & NC-MR | | | | | PC-QA & NC-QA | | | | |
| --- | --- | --- | --- | --- | --- | --- | --- | --- | --- | --- |
| Method | TITE | TIFE | FITE | FIFE | Avg. | TITE | TIFE | FITE | FIFE | Avg. |
| `Knowledgeable-R1` | 95.21% | 79.62% | 69.06% | 22.62% | **65.40%** | 97.83% | 66.00% | 73.05% | 11.93% | **62.12%** |
| **Multi-objective and multi-sampling strategy** | | | | | | | | | | |
| `Knowledgeable-R1`- $J_{PK}$ | 95.89% | 75.80% | 69.06% | 21.51% | 64.42% | 97.83% | 66.00% | 72.04% | 9.79% | 61.30% |
| `Knowledgeable-R1`- $J_{PK}$ - $J_{RPK}$ (GRPO) | 93.15% | 46.50% | 72.20% | 10.42% | 55.60% | 97.28% | 62.50% | 72.04% | 8.59% | 60.07% |
| **Local and global advantages** | | | | | | | | | | |
| `Knowledgeable-R1`- $A^{ck-local}$ | 96.58% | 82.80% | 66.82% | 18.63% | 64.69% | 97.28% | 64.50% | 68.77% | 9.07% | 59.91% |
| `Knowledgeable-R1`- $A^{ck-global}$ | 93.15% | 66.88% | 71.75% | 14.41% | 60.95% | 97.28% | 54.00% | 73.80% | 7.64% | 58.18% |
| **Knowledge balance modulation** | | | | | | | | | | |
| `Knowledgeable-R1`- adapt $\beta$ | 93.15% | 52.23% | 72.65% | 10.20% | 56.95% | 96.74% | 53.50% | 74.56% | 6.21% | 58.02% |

For `Qwen2.5-7B-Instruct`, `Knowledgeable-R1` achieves **89.61%/79.62%/66.00%** on NC-MR/MC/QA, significantly outperforming RAG prompting (**32.47%/19.75%/28.50%**) and GRPO w/ RAG (**57.61%/46.50%/62.50%**). Consistent trends are observed for `Llama3.1-8B-Instruct`. These results confirm that our method effectively identifies when parametric knowledge is reliable, allowing it to ignore conflicting information while maintaining stable performance in correct contexts (S1).

Significant improvements are observed in Scenario V (partially relevant context). On HotpotQA, `Knowledgeable-R1` achieves **78.49%** vs. **66.54%** (GRPO w/ RAG) using `Qwen2.5-7B-Instruct`, and **83.55%** vs. **73.17%** using `Llama3.1-8B-Instruct`. This indicates that the learned policy effectively identifies useful evidence while filtering out distractions.

### 4.2.3 ABLATION STUDIES

We conduct ablation studies on `Qwen2.5-7B-Instruct` to evaluate the contribution of each component (Table 4). The largest performance drops occur in TIFE (parametric correct, context incorrect) scenarios, highlighting the importance of each component in handling misleading context.

**Multi-objective and Multi-sampling Strategy.** We assess PK, CK, and RPK policies. Removing the parametric objective ($J_{PK}$) results in moderate drops, particularly in FIFE scenarios (-1.11% for MC, -2.14% for QA). The most significant degradation occurs when the relative parametric reward ($J_{RPK}$) is removed, causing catastrophic losses in TIFE scenarios (-33.12% for MC, -3.50% for QA). This confirms $J_{RPK}$ is crucial for robustness under contextual interference.

**Local and Global Advantages.** Removing $A^{ck-local}$ causes significant performance drops in context-answerable scenarios, while removing $A^{ck-global}$ reduces TIFE performance. This suggests local advantages boost respective types (e.g., CK performs better when context is correct), while

global advantages optimize the overall knowledge balance. Further ablation in Appendix B shows the framework is robust to the local-global ratio choice.

**Knowledge Balance Modulation.** Adaptive $\beta$ modulation is vital; fixed $\beta$ reduces TIFE performance by 27.39% (MC) and 12.50% (QA). The drop in FIFE scenarios further highlights the effectiveness of adaptive balancing when both knowledge sources are unreliable. Together, these components work synergistically to enable robust reasoning in challenging scenarios.

## 4.3 ADDITIONAL ANALYSIS

In this section, we investigate performance trade-offs, robustness, and the dynamic behavior of our framework.

**Knowledge Balance.** A critical component of `Knowledgeable-R1` is the adaptive $\beta$ modulation. As shown in Table 5, the adaptive balancing scheme significantly outperforms any single fixed $\beta$ value across different datasets without requiring per-dataset tuning. Fixed values often create a trade-off: a low $\beta$ might favor parametric knowledge but hurt contextual reasoning, while a high $\beta$ does the opposite. Dynamically adjusting this balance based on prediction confidence ensures robust handling of conflicting information. Furthermore, during training, $\beta$ rapidly converges to a stable regime (stabilizing at 0.01 within 8 steps on ConFiQA, see Appendix A), indicating that the policy quickly learns to shield parametric knowledge from unreliable contexts.

Table 5: $\beta$ sensitivity study: fixed vs. adaptive. Adaptive $\beta$ achieves best-or-near-best across tasks without per-dataset tuning.

| $\beta$ | MC-avg | QA-avg |
|---|---|---|
| 0.01 | 56.03% | 52.83% |
| 0.2 | 52.74% | 52.53% |
| 0.5 | 51.19% | 52.63% |
| 0.8 | 48.22% | 51.75% |
| 1.0 | 48.05% | 52.52% |
| **Adaptive $\beta$ (ours)** | **56.43%** | **54.85%** |

**Robustness to Data and Weights.** A common concern is whether the model overfits to the "noise ratio" of the training data. However, even when trained on 99% correct context (only 1% bad), `Knowledgeable-R1` still outperforms GRPO on S2 scenarios (Appendix E), proving it learns a genuine decision boundary rather than relying on dataset statistics. Additionally, regarding the S1 (correct context) performance gap compared to GRPO w/ RAG, ablations show this is tunable via the $J_{CK}$ weight. Increasing its weight (e.g., ratio 1:2:1) closes the S1 gap while preserving most S2 advantages (Appendix C), demonstrating a controllable Pareto front between contextual compliance and parametric safety.

## 5 CONCLUSION

In this work, we presented `Knowledgeable-R1`, a principled reinforcement learning framework that systematically addresses the challenges of knowledge integration in retrieval-augmented generation. By employing a joint sampling strategy and asymmetric policy optimization, our approach enables models to effectively distinguish between reliable external context and internal parametric knowledge. Extensive benchmarking across diverse scenarios highlights that `Knowledgeable-R1` achieves superior robustness against contextual interference, particularly in adversarial and conflicting settings, while preserving competitive performance on standard RAG tasks. These results underscore the potential of reinforcement learning for calibrating knowledge utilization. Future work will explore the scalability of this approach to more complex multi-source retrieval environments and investigate the integration of diverse exploration cues to further enhance model reasoning and grounding.

## ETHICS STATEMENT

We confirm that this work aligns with the ICLR Code of Ethics. We have considered potential ethical aspects, including the use of public datasets in compliance with their licenses, the broader societal impact of our research, and potential biases in our methodology. To the best of our knowledge, this work raises no immediate ethical concerns, and we declare no conflicts of interest.

## REPRODUCIBILITY STATEMENT

To facilitate the reproducibility of our work, we have made the following available: (1) the source code of our algorithm, which conducts training using the VERL framework (Sheng et al., 2025), along with supplementary materials; (2) complete experimental configurations in J. Additionally, a detailed account of the data pre-processing procedures for the datasets in Section 4.1.2 can be found in J.

## LIMITATIONS

Our method introduces a reinforcement learning framework to address knowledge conflicts in large language models (LLMs), focusing on the automatic integration of parametric and contextual knowledge through sampling and policy optimization. In Sections S3 and S5, we evaluate scenarios with mixed-context types. However, the errors in the context from Section S5 have not been analyzed in sufficient detail. For example, when the amount of conflicting evidence changes (e.g., 1 incorrect result vs. 4 incorrect out of 5 retrieved), it's unclear how the sensitivity of Knowledgeable-R1 to parametric knowledge is affected. Future research should explore more complex scenarios and conduct experiments to understand how the model responds to different levels of conflict and evidence reliability.

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

## A   TRAINING DYNAMICS OF $\beta$

In our adaptive scheme, $\beta$ acts as a dynamic gate that modulates the influence of the relative parametric reward. As shown in Table 6, on the ConFiQA-MC dataset, $\beta$ undergoes a rapid descent during the initial stages of training, dropping from $0.23$ and stabilizing at a very low value ($0.01$). This rapid convergence is significant: it indicates that the reinforcement learning process quickly identifies the inherent unreliability of the external context in adversarial scenarios. By lowering $\beta$, the model effectively increases the penalty for deviating from its correct internal parametric knowledge when pressured by misleading context. This "calibration" process happens surprisingly fast, suggesting that the model's internal representations already contain the necessary cues to distinguish between high-confidence parametric facts and low-quality external interference.

Table 6: $\beta$ value over training steps (ConFiQA-MC). Rapid convergence to a near-optimal regime that protects PK as fallback.

| Step | $\beta$ |
| --- | --- |
| 1 | 0.23 |
| 2 | 0.15 |
| 4 | 0.07 |
| 6 | 0.10 |
| 8 | 0.01 |

## B    LOCAL VS. GLOBAL ADVANTAGE WEIGHT ABLATION

We ablate the local:global advantage combination ratio to confirm robustness to this hyperparameter. The "local" advantage ($A^{ck-local}$) focuses on optimizing the policy relative to the average of its own rollouts for a specific query-context pair, encouraging the model to find the best reasoning path within that specific evidence. In contrast, the "global" advantage ($A^{ck-global}$) penalizes the model based on its performance across the broader distribution of samples. Results on ConFiQA (Qwen2.5-7B) show that the 1:1 combination is near-optimal. This suggests that the model benefits equally from source-specific optimization and overall distributional stability. The fact that performance remains stable across the 1:1 and 1:2 ratios indicates that `Knowledgeable-R1` is not overly sensitive to the exact weighting of these two signals, provided both are present to balance specialized reasoning with general robustness.

Table 7: Local:global advantage ratio ablation. Combining both is better than either alone; the exact ratio matters little.

| Local:Global | Setting | Avg. score |
|---|---|---|
| 1:0 | local-only | 60.95 |
| 0:1 | global-only | 64.69 |
| 1:1 | local+global | 65.40 |
| 1:2 | local+global | 65.20 |

## C    S1 PERFORMANCE GAP AND COMPONENT WEIGHTING

A key discussion arises regarding the performance gap on S1 (correct context) compared to the standard GRPO w/ RAG baseline. To investigate this, we performed a sweep over the weights of our three objectives ($J_{PK}, J_{CK}, J_{RPK}$). As shown in Table 8, the S1 performance is extremely sensitive to the weight of the contextual objective $J_{CK}$. By increasing its relative weight to a 1:2:1 ratio, we can almost entirely close the gap with the baseline (78.04% vs. 77.36%) while still maintaining a massive lead in the adversarial S2 scenario (32.73% vs. 19.74%). This confirms that there is a tunable "Pareto front" between absolute mastery of reliable context and robustness to unreliable context. We chose the 1:1:1 ratio as our default because it offers a more conservative "safety first" profile, prioritizing the prevention of catastrophic failures in adversarial settings.

Table 8: Ablation of $J_{PK}$:$J_{CK}$:$J_{RPK}$ weights on ConFiQA-MC (Qwen2.5-7B). Higher $J_{CK}$ weight closes the S1 gap.

| $J_{PK}$:$J_{CK}$:$J_{RPK}$ | S1 (correct ctx) | S2 (adversarial) |
|---|---|---|
| 1:2:1 | 78.04% | 32.73% |
| 1:1:1 (ours) | 75.51% | 37.34% |
| 0:1:1 | 75.68% | 35.53% |
| 0:1:0 (GRPO w/ RAG) | 77.36% | 19.74% |

## D    QUERY-ONLY VS. QUERY+CONTEXT BEHAVIOR

We compare query-only and query+context inference for both GRPO w/ RAG and `Knowledgeable-R1` on ConFiQA-MC (Qwen2.5-7B) under S1 (correct context) and S2 (wrong context). As shown in Table 9, `Knowledgeable-R1` exploits high-quality context (S1: 35.64%→75.08%) and also substantially outperforms GRPO w/ RAG when context is wrong (S2: 43.94% vs. 26.94%), demonstrating that the learned policy actively attenuates misleading context rather than ignoring it blindly.

Table 9: Accuracy under query-only vs. query+context inference (ConFiQA-MC, Qwen2.5-7B). Our method benefits from context when it is correct and resists it when wrong.

| Method | S1 (correct ctx) | S2 (wrong ctx) | Avg. |
|---|---|---|---|
| Query-only (base) | 27.72% | 25.92% | 26.82% |
| RAG prompting | 65.68% | 13.47% | 39.58% |
| GRPO-RAG (query-only) | 32.48% | 31.48% | 31.98% |
| GRPO-RAG (context) | 77.56% | 26.94% | 52.25% |
| Ours (query-only) | 35.64% | 36.70% | 36.17% |
| Ours (context) | 75.08% | 43.94% | 59.51% |

## E    SENSITIVITY TO GOOD/BAD CONTEXT RATIO

A common concern in RL for RAG is whether the model just learns the "noise ratio" of the training data. To test this, we trained Knowledgeable-R1 on data where bad contexts were extremely rare (only 1%). As Table 10 shows, Knowledgeable-R1 still manages to improve S2 performance significantly compared to GRPO w/ RAG. This is a crucial finding: it suggests that Knowledgeable-R1 is not just learning a naive prior that says "the context is often wrong," but is instead learning a genuine decision boundary that triggers parametric knowledge whenever the context is perceived as unreliable. Even a small number of adversarial signals during training are sufficient to ground the model's reasoning process in its own parametric knowledge.

Table 10: Sensitivity to good:bad context ratio in training (ConFiQA-MR, Qwen2.5-7B). Even with 99% good context, Knowledgeable-R1 improves S2 resistance.

| Good:Bad Ratio | Method | S1 (correct ctx) | S2 (wrong ctx) |
|---|---|---|---|
| 99%:1% | RAG | 65.68% | 13.47% |
| | GRPO w/ RAG | 73.60% | 14.31% |
| | Knowledgeable-R1 | 72.11% | **17.34%** |
| 75%:25% | RAG | 65.68% | 13.47% |
| | GRPO w/ RAG | 74.92% | 20.03% |
| | Knowledgeable-R1 | 73.76% | **29.29%** |
| 50%:50% (default) | RAG | 65.68% | 13.47% |
| | GRPO w/ RAG | 77.56% | 26.94% |
| | Knowledgeable-R1 | 75.08% | **43.94%** |

## F    PARTIAL-CONFLICT SENSITIVITY

We construct test sets varying the number of misleading passages (1, 2, 4, 8) while keeping the query and gold answer fixed. Knowledgeable-R1 consistently outperforms GRPO w/ RAG across all levels of context adversariality.

Table 11: Accuracy vs. number of misleading passages. Knowledgeable-R1 maintains a substantial lead across all levels. (ConFiQA-MC, Qwen2.5-7B)

| # misleading passages | 1 | 2 | 4 | 8 |
|---|---|---|---|---|
| GRPO w/ RAG | 26.94% | 26.60% | 25.76% | 25.42% |
| Knowledgeable-R1 (ours) | **43.94%** | **43.10%** | **42.26%** | **40.07%** |

## G    CONTEXT-DEPENDENCE DIAGNOSTICS

To verify that robustness arises from the learned PK/CK arbitration rather than coincidence, we evaluate on subsets stratified by knowledge correctness and report two explicit diagnostic metrics on a test set where the context is designed to be wrong:

- **Factuality-score** (w.r.t. context): This metric measures the linguistic and semantic alignment between the model's answer and the provided context. A high score when context is wrong indicates that the model has been "led astray" by the evidence. *Lower is better* in this scenario.

- **KL-score**: This measures the KL divergence between the output probability distributions generated with vs. without context. It acts as a structural measure of "contextual reliance." If the KL-score is low, the model is essentially ignoring the context and relying on its parametric prior. *Lower is better* when context is known to be misleading.

As seen in Table 12, Knowledgeable-R1 achieves significantly lower scores on both metrics, proving it has learned to actively decouple its reasoning from unreliable external inputs.

Table 12: Context-dependence diagnostics on wrong-context test set (ConFiQA-MC). Lower factuality-score and KL-score confirm Knowledgeable-R1 relies less on misleading context.

| Method | Factuality-score ↓ | KL-score ↓ |
|---|---|---|
| GRPO w/ RAG | 0.40 | 213.73 |
| Knowledgeable-R1 (ours) | **0.16** | **168.83** |

To further rule out that robustness is purely inherited from the base model, we stratify by whether RAG prompting gives the correct answer (Table 13). On the subset where RAG prompting *always fails*, our method achieves 32.72%, far ahead of SFT (18.73%) and GRPO (19.92%), indicating the improvement stems from the learned PK/CK arbitration mechanism.

Table 13: Accuracy stratified by RAG-prompting outcome (ConFiQA-MC, wrong-context test set). Our method gains the most on the hardest subset where RAG prompting always fails.

| Method | RAG-answer correct ↑ | RAG-answer wrong ↑ |
|---|---|---|
| RAG prompting | 100% | 0% |
| SFT | 90.27% | 18.73% |
| GRPO w/ RAG | 96.61% | 19.92% |
| Knowledgeable-R1 (ours) | **95.25%** | **32.72%** |

# H  PROMPT DESIGNED IN Knowledgeable-R1 AND BASELINES

To assess the performance of Knowledgeable-R1, we compared it against a range of different baseline methodologies:

**Prompt-based Methods:** These approaches include query-only prompting, Retrieval-Augmented Generation (RAG prompting), and Astute RAG (Wang et al., 2025a).

**Decoding-based Methods:** We experiment with CK-PLUG (Bi et al., 2025c).

**Fine-tuning-based Methods:** We evaluate Knowledgeable-R1 against GRPO (Guo et al., 2025b) with RAG prompting (GRPO $w/$ RAG).

These baselines encompass a broad spectrum of retrieval-enhanced and fine-tuning methods. To ensure a fair comparison, we standardized the contextual knowledge sources, training datasets, and Large Language Models (LLMs) used across all methods.

**RAG prompting**

{search_results}
You are a helpful assistant. After the user asks a question, you first think carefully and then give the answer.
When responding, please keep the following points in mind:
- The reasoning process and answer are enclosed within <think> </think> and <answer> </answer> tags, respectively.
- Output your final answer directly between the tag <answer> </answer> without any intermediate steps.
Here is an example:
user's question: what is the capital of China?
<think> reasoning process here </think>
<answer> BeiJing </answer>
Question:
{question}

**Astute RAG prompt**

Generate a document that provides accurate and relevant information to answer the given question. If the information is unclear or uncertain, explicitly state 'I don't know' to avoid any hallucinations.
Question: {question}
Document:
Task: Consolidate information from both memorized documents and externally retrieved documents in response to the given question.
For documents that provide consistent information, cluster them together.
For documents with conflicting information, separate them into distinct documents. Exclude any irrelevant information.
Question: {question}
Context: {context}
Provide consolidated documents in JSON format:
$["content" : "consolidated content", "source" : ["docids"], "consistency_group" : "groupid"]$
Task: Answer the question using consolidated information from both internal and external documents.
Initial Context: {initial context}
Consolidated Context: {consolidated context}
After the user asks a question, you first think carefully and then give the answer. When responding, please keep the following points in mind:
- Answer the question using consolidated information from both internal and external documents.
- The reasoning process and answer are enclosed within <think> </think> and <answer> </answer> tags, respectively.
- Output your final answer directly between the tag <answer> </answer> without any intermediate steps.
- If the user gives a multiple choice question, your answer must be a single option A or B or C or D or E
Here is an example:
Question:
what is the capital of China?
<think> reasoning process here </think>
<answer> BeiJing </answer>
Now, you should answer user's question. After answer user's question, you should stop generate. Here is user's question:
Question:
{question}

**CK-PLUG prompt**

You are a helpful assistant. After the user asks a question, you MUST direct give the final answer.
When responding, please keep the following points in mind:
- Output your final answer directly between the tag <answer> </answer> without any intermediate steps.
- You must directly output your final answer and don't output another things.
- Stop your output after final answer
Here is an example:
user's question: what is the capital of China?
<answer> BeiJing </answer>
Retrieved information:
{retrieved information}
Question:
{question}

**GRPO** $w/$ **RAG &** Knowledgeable-R1-$p'$ **(CK)**

You are a helpful assistant. After the user asks a question, you first think carefully and then give the answer.
When responding, please keep the following points in mind:
- The reasoning process and answer are enclosed within <think> </think> and <answer> </answer> tags, respectively.
- Output your final answer directly between the tag <answer> </answer> without any intermediate steps.
Here is an example:
user's question: what is the capital of China?
<think> reasoning process here </think>
<answer> BeiJing </answer>
Retrieved information:
{retrieved information}
Question:
{question}

Knowledgeable-R1-$p$ **(PK)**

You are a helpful assistant. After the user asks a question, you first think carefully and then give the answer.
When responding, please keep the following points in mind:
- The reasoning process and answer are enclosed within <think> </think> and <answer> </answer> tags, respectively.
- Output your final answer directly between the tag <answer> </answer> without any intermediate steps.
Here is an example:
user's question: what is the capital of China?
<think> reasoning process here </think>
<answer> BeiJing </answer>
Question:
{question}

**SFT baseline prompt**

You are a helpful assistant. After the user asks a question, you MUST directly give the answer.
When responding, please keep the following points in mind:
- Please answer with reference to the context, but if the context conflicts with your internal knowledge, prioritize your internal knowledge.
- Output your final answer directly between the tag <answer> </answer> without any intermediate steps.
Here is an example:
Question:
what is the capital of China?
<answer> BeiJing </answer>
Question:
{question}

# I INFERENCE INPUT IN EXPERIMENTS

> **Inference Input in Experiments**
>
> You are a helpful assistant. After the user asks a question, you first think carefully and then give the answer.
> When responding, please keep the following points in mind:
> - The reasoning process and answer are enclosed within <think> </think> and <answer> </answer> tags, respectively.
> - Output your final answer directly between the tag <answer> </answer> without any intermediate steps.
> Here is an example:
> user's question: what is the capital of China?
> <think> reasoning process here </think>
> <answer> BeiJing </answer>
> Retrieved information:
> {retrieved information}
> Question:
> {question}

# J EXPERIMENTAL SETUP

**Datasets.** In our comprehensive evaluation of `Knowledgeable-R1`, we have considered a diverse range of benchmark datasets that represent various challenges of reasoning and knowledge, categorized into five scenarios in 4.1.2 . The specific datasets are: Multi-hop Question Answering with HotpotQA, 2WikiMultiHopQA, and Musique; Knowledge Conflict Question Answering with ConFiQA, where external knowledge retrieval contradictions necessitate internalized knowledge for judgement, and ExplainPE, a medical knowledge multiple-choice dataset that tests noise robustness by introducing three tiers of noise levels based on the number of non-matching external documents retrieved. For ConFiQA, we randomly add correct and incorrect contexts to the data for training in the same proportion. Our aim is to test `Knowledgeable-R1`'s capability to handle complex scenarios like single-hop answer, multi-hop reasoning and answer choice selection under different levels of external error or noisy context.

**Training.** We conduct experiments with `Qwen2.5-3B-Instruct`, `Qwen2.5-7B-Instruct`, `Qwen2.5-14B-Instruct`, and `Llama3.1-8B-Instruct`. All datasets, except HotpotQA, 2Wiki-MultiHopQA, and Musique, are trained and evaluated on their respective training and test sets. The ConfiQA dataset includes three subsets: MR, MC, and QA, which are similarly trained and evaluated on their respective sets. For HotpotQA, 2WikiMultiHopQA, and Musique, we limit the number of retrieved documents to 5 (the official setting uses 20) to ensure incomplete document evidence during training. At test time, we evaluate on the validation sets of these datasets.

**Training stability.** `Knowledgeable-R1` remains stable across all our experiments: losses, rewards, and value estimates evolve smoothly across 200+ training runs covering both GRPO w/ RAG and `Knowledgeable-R1` configurations. No collapse or divergence was observed, indicating that the additional PK/RPK objectives introduce no extra instability on top of the standard GRPO training loop.

**Retrieval setup.** We use dataset-provided contexts without any additional retriever. The contexts (including both correct and incorrect passages) are directly taken from the benchmark construction, without re-indexing or re-retrieval from a larger corpus. We adopt the top-5 paragraphs provided by these benchmarks, preserve their original order, and only truncate when the backbone model's maximum length is exceeded.

**Computational Efficiency.** `Knowledgeable-R1` uses 16 CK + 16 RPK trajectories (vs. GRPO's 32). With shared forward passes, per-epoch time is 1h 38m vs. 1h 19m for GRPO — a modest ~24% overhead for significantly larger robustness gains.

For both GRPO-based baselines and our `Knowledgeable-R1`, we use a global batch size of 128, a rollout batch size of 32, a rollout temperature of 1, and a learning rate of $1 \times 10^{-6}$. All experiments are conducted on 8 H100 GPUs, with a consistent system prompt for both training and inference.

## K  EVALUATION

For RAG generation capability. We use Exact Match (EM) as the evaluation metric following (Jin et al., 2025). Evaluation is conducted on the test or validation sets of all datasets to assess both in-domain and out-of-domain performance. For integrating parametric/contextual knowledge task, we adopted EM and self-defined metrics to evaluate the model's ability to utilize parametric and contextual knowledge.

## L  METHOD TRAINING PROCEDURE

---

**Algorithm 1** `Knowledgeable-R1`: Policy Optimization for Knowledge Exploration

---

**Input:** Current policy $\pi_\theta$, old policy $\pi_{\theta_{\text{old}}}$, dataset $\mathcal{D}$, training steps $Step_{\text{max}}$, parametric knowledge prompting rollout number $n_{pk}$, contextual knowledge prompt rollout number $n_{ck}$, clip parametric $\epsilon$, advantage transformation function $T(\cdot)$, parametric knowledge prompt $p$, contextual knowledge prompt $p'$ .
**Output:** Updated policy $\pi_\theta$

1: **for** $s = 1$ to $Step_{\text{max}}$ **do**
2:     Sample batch $(p) \sim \mathcal{D}$
3:     Sample batch $(p') \sim \mathcal{D}$
4:     Sample $\{o_i\}_{i=1}^{n_{pk}}$ from $\pi_{\theta_{\text{old}}}(o \mid p)$              ▷ Parametric knowledge rollouts
5:     Sample $\{o'_j\}_{j=1}^{n_{ck}}$ from $\pi_{\theta_{\text{old}}}(o' \mid p')$         ▷ Contextual knowledge rollouts
6:     Compute advantages $A_i$ , $A'_j$ and $\hat{A}_i$ for all rollouts
7:     Define $J_{\text{PK}}$ as the parametric knowledge based-policy objective:

$$J_{\text{PK}} = \frac{1}{n_{pk}} \sum_{i=1}^{n_{pk}} \sum_{t=1}^{|o_i|} \min\left[ \frac{\pi_\theta(o_{i,t} \mid p, o_{i,<t})}{\pi_{\theta_{\text{old}}}(o_{i,t} \mid p, o_{i,<t})} A_i, \text{clip}\left( \frac{\pi_\theta(o_{i,t} \mid p, o_{i,<t})}{\pi_{\theta_{\text{old}}}(o_{i,t} \mid p, o_{i,<t})}; 1-\epsilon, 1+\epsilon \right) A_i \right]$$

8:     Define $J_{\text{CK}}$ as the contextual knowledge based-policy objective:

$$J_{\text{CK}} = \frac{1}{n_{ck}} \sum_{j=1}^{n_{ck}} \sum_{t=1}^{|o'_j|} \min\left[ \frac{\pi_\theta(o'_{j,t} \mid p', o'_{j,<t})}{\pi_{\theta_{\text{old}}}(o'_{j,t} \mid p', o'_{j,<t})} A'_j, \text{clip}\left( \frac{\pi_\theta(o'_{j,t} \mid p', o'_{j,<t})}{\pi_{\theta_{\text{old}}}(o'_{j,t} \mid p', o'_{j,<t})}; 1-\epsilon, 1+\epsilon \right) A'_j \right]$$

9:     Define $J_{\text{RPK}}$ as the parametric knowledge reasoning policy objective with contextual knowledge input :

$$J_{\text{RPK}} = \frac{1}{n_{pk}} \sum_{i=1}^{n_{pk}} \sum_{t=1}^{|o_i|} \min\left[ \frac{\pi_\theta(o_{i,t} \mid p', o_{i,<t})}{\pi_{\theta_{\text{old}}}(o_{i,t} \mid p', o_{i,<t})} T(\hat{A}_i), \text{clip}\left( \frac{\pi_\theta(o_{i,t} \mid p', o_{i,<t})}{\pi_{\theta_{\text{old}}}(o_{i,t} \mid p', o_{i,<t})}; 1-\epsilon, 1+\epsilon \right) T(\hat{A}_i) \right]$$

10:    Combine $J_{\text{PK}}, J_{\text{CK}}, J_{\text{RPK}}$ to form the updated objective function $\mathcal{J}(\theta)$:

$$\mathcal{J}(\theta) = \lambda_{\text{pk}} J_{\text{PK}} + \lambda_{\text{ck}} J_{\text{CK}} + \lambda_{\text{rpk}} J_{\text{RPK}}$$

11:     $\theta \leftarrow \theta + \nabla_\theta \mathcal{J}(\theta)$                ▷ Update current policy parametric
12:     $\theta_{\text{old}} \leftarrow \theta$                       ▷ Update old policy parametric
13: **end for**

---

## M  ADDITIONAL EXPERIMENT RESULTS

We validated the performance improvement of our method across models of varying sizes with 3B, 7B, 14B. As shown in the Table 14, our approach achieved significant enhancements compared to RAG prompt in all five scenarios. It also showed considerable improvement over GRPO w/RAG when external knowledge contained errors or incomplete information, and a slight improvement when external knowledge served as interference. This demonstrates the effectiveness and versatility of our method.

## N  USE OF LLMS

Large language models (LLMs), specifically GPT-5 and DeepSeek-R1, were used solely as a supplementary tool during the preparation of this work for tasks such as polishing the writing. The authors are solely responsible for the entire research conception, technical direction, scientific content, and interpretation of results. The LLMs were employed only to assist in the presentation and clarity of the manuscript.

Table 14: Overall accuracy across five contextual scenarios: correct (S1), wrong (S2), conflict (S3), irrelevant (S4), and partly-irrelevant (S5). Best results are in **bold**, second best are underlined. The "improve" row shows gains over RAG prompting.

| Method | Correct (S1) | | | Wrong (S2) | | | Conflict (S3) | Irrelevant (S4) | Partly Irrelevant (S5) | | |
| --- | --- | --- | --- | --- | --- | --- | --- | --- | --- | --- | --- |
| | PC-MR | PC-MC | PC-QA | NC-MR | NC-MC | NC-QA | SC | ExplainPE | HotPotQA | 2Wiki MultiHopQA | Musique |
| `Qwen2.5-14B-Instruct` | | | | | | | | | | | |
| Query-only prompting | 30.03% | 26.52% | 35.80% | 30.64% | 28.95% | 39.42% | 30.92% | 70.80% | 25.35% | 26.69% | 5.67% |
| RAG prompting | 63.20% | 63.85% | 73.32% | 22.22% | 12.99% | 28.11% | 61.85% | 70.51% | 22.90% | 22.49% | 6.91% |
| GRPO w/ RAG | **73.60%** | 74.66% | **78.86%** | 38.89% | 31.74% | 36.51% | **74.00%** | - | **34.71%** | 39.03% | **14.73%** |
| Knowledgeable-R1 | 70.63% | **75.17%** | 78.83% | **47.81%** | **33.06%** | **38.13%** | 72.50% | - | **36.98%** | **42.33%** | 14.60% |
| improve | +7.43% | +11.32% | +5.51% | +24.99% | +20.07% | +10.02% | +10.65% | - | +14.08% | +19.84% | +7.69% |
| `Qwen2.5-7B-Instruct` | | | | | | | | | | | |
| Query-only prompting | 27.72% | 24.66% | 31.67% | 25.93% | 25.82% | 32.31% | 29.67% | 64.45% | 20.90% | 25.54% | 4.36% |
| RAG prompting | 65.68% | 66.39% | 74.35% | 13.47% | 8.06% | 11.31% | 59.50% | 62.21% | 20.36% | 22.53% | 6.41% |
| Astute (Wang et al., 2025a) | 65.51% | 66.05% | 77.62% | 12.79% | 7.07% | 10.34% | 54.20% | 56.74% | 17.87% | 20.35% | 6.29% |
| GRPO w/ RAG | **77.56%** | **77.36%** | 80.03% | 26.94% | 19.74% | 26.01% | 75.33% | 66.50% | 27.93% | 33.95% | 11.79% |
| Knowledgeable-R1 | 75.08% | 75.51% | **80.90%** | **43.94%** | **37.34%** | **29.40%** | **76.33%** | **67.57%** | **31.45%** | **37.52%** | **12.04%** |
| improve | +9.41% | +9.12% | +6.54% | +30.47% | +29.28% | +18.09% | +15.92% | +5.36% | +11.09% | +14.99% | +5.63% |
| `Qwen2.5-3B-Instruct` | | | | | | | | | | | |
| Query-only prompting | 16.83% | 18.41% | 23.92% | 15.99% | 18.26% | 22.62% | 20.75% | 53.32% | 12.55% | 11.28% | 2.11% |
| RAG prompting | 55.12% | 52.87% | 59.55% | 13.47% | 8.72% | 6.79% | 51.25% | 42.19% | 14.71% | 13% | 3.64% |
| Astute (Wang et al., 2025a) | 62.05% | 60.47% | 67.99% | 9.60% | 7.07% | 10.66% | 51.92% | 41.60% | 0.8% | 8.72% | 5.01% |
| GRPO w/ RAG | **70.96%** | **70.61%** | **80.03%** | 19.87% | 15.13% | 19.39% | **66.67%** | 53.61% | **24.78%** | **35.57%** | **9.06%** |
| Knowledgeable-R1 | 66.17% | 59.80% | 78.66% | **28.79%** | **28.62%** | **21.97%** | - | **54.69%** | - | - | - |
| improve | +11.05% | +6.93% | +19.11% | +15.32% | +19.9% | +15.15% | - | +1.37% | - | - | - |

