# OpenReview forum: "Resisting Contextual Interference in RAG via Parametric-Knowledge Reinforcement"
_ICLR.cc/2026/Conference — ICLR 2026 Poster_

### Official Review · Reviewer_C4dV · 2025-10-26

**Soundness:** 4
**Presentation:** 3
**Contribution:** 4
**Rating:** 6
**Confidence:** 4

**Summary:**

To address the tendency of models to disregard **parametric knowledge (PK)** when the retrieved context in retrieval-augmented generation is **noisy, misleading, or contradictory**, the authors propose **Knowledgeable-R1** — a **reinforcement learning (RL)** framework that explicitly trains LLMs to resist contextual interference while still leveraging useful retrieved information.

Training follows a **GRPO-style** setup with three policies:

* **PK:** query-only (parametric knowledge)
* **CK:** query + context (context-aware)
* **RPK:** query + context, but decoding along the PK trajectory to test robustness

**Strengths:**

**Well-motivated problem.** The work addresses a practical and underexplored limitation of RAG systems — their lack of robustness to noisy contexts.

**Strong empirical validation.** Results hold across two model families, four model sizes, and five datasets, showing consistent robustness gains under four interference conditions while preserving accuracy on clean contexts.

**Comprehensive ablations.** Each design choice (joint sampling, local/global advantages, adaptive β) is empirically justified, with clear performance drops when omitted.

**Novelty.** Unlike prior approaches that dealt with noisy contexts (e.g., [1], [2], [3]), this framework explicitly operationalizes the PK–CK trade-off in a reinforcement learning manner rather than using prompt engineering or fine-tuning. It closely resembles [4], although that work is quite recent and may not have been available when this paper was written.

[1] Asai, Akari, et al. "Self-rag: Self-reflective retrieval augmented generation." NeurIPS 2023 workshop on instruction tuning and instruction following. 2023.

[2] Yoran, Ori, et al. "Making retrieval-augmented language models robust to irrelevant context." arXiv preprint arXiv:2310.01558 (2023).

[3] Wang, Fei, et al. "Astute rag: Overcoming imperfect retrieval augmentation and knowledge conflicts for large language models." arXiv preprint arXiv:2410.07176 (2024).

[4] Xu, Tingqiao, et al. "DyKnow-RAG: Dynamic Knowledge Utilization Reinforcement Framework for Noisy Retrieval-Augmented Generation in E-commerce Search Relevance." arXiv preprint arXiv:2510.11122 (2025).

**Weaknesses:**

**Key method part is not well-explained**. It is not quite clear for me how RPK sampling works. in line 152 RPK policy is defined as $ \pi (o_t \mid p^{\prime}, o_{<t})$ - output is conditioned on previously generated tokens $o_{<t}$ and $p^{\prime} =$ query + context. But in the next paragraph, the authors say that $o_t$ is conditioned on query-only input  $p$: "we use $o$ to denote a token sequence generated when the current policy is conditioned on $p$ (query-only input)". Then the authors say that $o$ means possibly two different things - output of PK ("answer from parametric knowledge") and output of RPK ("answer consistent with PK"). Could you please elaborate what is the difference between these two and maybe use different variables for them? What is input and output when each new token in RPK is generated? I think it is very important as RPK is one of the key differences of Knowledgeable-R1 from GRPO + RAG baseline.

**Heuristic nature of β-modulation.** The adaptive penalty scaling is empirically tuned, with limited theoretical justification. A more formal grounding or ablation across β dynamics would strengthen the argument.

**Limited exploration of partial-conflict granularity.** The paper notes this in its limitations — sensitivity analysis on varying proportions of misleading passages would be valuable.

**Narrow task domain.** Evaluation focuses on QA-style tasks; it remains unclear whether the learned robustness generalizes to open-ended generation, dialogue, or summarization settings.

**Confusing writing:**

- Baselines are not well explained in Section 4.1.1:
  - CK-Plug is omitted.
  - GRPO with RAG is not clearly described — the difference from Knowledgeable-R1 is unclear.
  - Key explanations appear only in the appendix.
- Lines 187–209 — indices *i*, *j*, *k* are unclear. Do they denote rollout indices?
  - The distinction between them is not specified — could they be unified under a single index?
- Font size in Tables 2 and 3 is too small to be properly readable.

**Typos**
- In Figure 1, the text “Probability distribution” overlaps with the image border.

**Questions:**

- Why don't you include KL-regularizer $\lambda_{KL} D_{KL}[\pi_\theta \| \pi_{ref}]$ in your total objective to avoid drifting too far from the reference policy, same as in [1], for example?
- What does the barplot in the top-right corner (the one with values 37.3, 19.7, 8.1 and 25.8) of Figure 1 mean?

[1] Xu, Tingqiao, et al. "DyKnow-RAG: Dynamic Knowledge Utilization Reinforcement Framework for Noisy Retrieval-Augmented Generation in E-commerce Search Relevance." arXiv preprint arXiv:2510.11122 (2025).

---

> ### Author Response · Authors · 2025-11-28
> **Reponse for C4dV - Weakness 1-4**
>
> Thank you very much for your thoughtful review and for recognizing both the importance of robustness to noisy contexts and the strengths of our framework (problem motivation, broad empirical validation, comprehensive ablations, and RL-based PK–CK arbitration). Below we address your concerns point by point.
>
> ---
>
>
> **Response to Weakness 1. RPK sampling and notation**
>
> **RPK uses query+context as input but is trained to reproduce the PK answer, so the RPK and PK outputs are the same sequence.**
> Concretely, for each query we first obtain a PK answer by sampling a trajectory
> $(o^{\text{PK}} = (o_1^{\text{PK}},\dots,o_T^{\text{PK}}))$ from the current policy conditioned on the query-only prompt (p), i.e.
> $(\pi_\theta(\cdot \mid p, o_{<t}^{\text{PK}}))$.
> In the RPK branch, we then **reuse this same sequence** as the target answer, but feed the model the  prompt (p' =) (query + context). At each decoding step (t), the policy computes
> $(\pi_\theta(\cdot \mid p', o^{\text{PK}}_{<t}))$
> and we maximize the log-probability of the PK token $(o^{\text{PK}}_t) $under the RPK reward. In other words, RPK does not generate an independent “RPK answer”, it re-evaluates the PK answer under query+context inputs and encourages the model to keep preferring the PK tokens when the context is misleading.
>
> ---
>
>
> **Response to Weakness 2. Heuristic nature of β-modulation**
>
> **Adaptive β consistently outperforms all fixed settings and approximates the best per-dataset choice.**
>
> We conduct a sensitivity study on ConFiQA-MC, ConFiQA-QA and SC datasets by fixing β and comparing to the adaptive scheme:
>
> |β|MC-avg|QA-avg|SC|
> |-|-|-|-|
> |**0.01**|56.03% (best fixed)|52.83% (best fixed)|73.91%|
> |**0.2**|52.74%|52.53%|74.66% (best fixed)|
> |**0.5**|51.19%|52.63%|73.5%|
> |**0.8**|48.22%|51.75%|73.16%|
> |**1.0**|48.05%|52.52%|72.15%|
> |**Adaptive β**|**56.43%**|**54.85%**|**76.33%**|
>
> Different datasets have different optimal beta values, while our adaptive beta achieves an excellent balance and delivers the best performance.
>
> ---
>
>
> **Response to Weakness 3. Limited exploration of partial-conflict granularity**
>
> **We provide a sensitivity analysis over different numbers of misleading passages and observe consistent gains for our method.**
> To partially address this concern, we construct test sets with 1, 2, 4, and 8 misleading passages while keeping the query and gold answer fixed. The accuracy of different methods is:
>
> |misleading passages|1|2|4|8|
> |-|-|-|-|-|
> |GRPO W/ RAG|26.94%|26.60%|25.76%|25.42%|
> |Ours|**43.94%**|**43.10%**|**42.26%**|**40.07%**|
>
> ---
>
>
> **Response to Weakness 4. Narrow task domain (QA focus)**
>
> **We deliberately focus on QA-style noisy-RAG benchmarks to study knowledge conflicts in a controlled and comparable setting, and we view broader open-ended tasks as important but out of scope for this submission.**
> Knowledge conflicts between parametric and retrieved knowledge are most commonly studied on QA-style datasets, where answer correctness is well-defined and context quality can be precisely controlled.  also primarily evaluates on QA-style settings when analyzing robustness to imperfect retrieval and parametric–contextual trade-offs.
>
> To keep the scope manageable and comparable to prior work such as CK-PLUG [1] and Astute RAG [2], we center our main evaluation on QA-style benchmarks that explicitly instantiate conflicting or misleading passages. Due to the current lack of publicly available datasets focused on knowledge conflicts in open-domain settings, we plan to address this issue in future work.
>
> [1]: Parameters vs. context: Fine-grained control of knowledge reliance in language models
>
> [2]: Astute rag: Overcoming imperfect retrieval augmentation and knowledge conflicts for large language models
>
> ---

---

> > ### Author Response · Authors · 2025-11-28
> > **Reponse for C4dV - Weakness 5-6 & Question 1-2**
> >
> > **Response to Weakness 5. Clarity of baselines (CK-Plug, GRPO w/ RAG, and main-text vs. appendix)**
> >
> > * **CK-PLUG.** We will add a short description in Section 4.1.1: CK-PLUG is a plug-and-play method that detects knowledge conflicts via “confidence gain” and then adjusts the token probabilities of negative-confidence-gain spans using a single control parameter, enabling fine-grained control over parametric vs. contextual reliance. [1]
> >
> > * **GRPO with RAG vs. our method.**  We will clarify that “GRPO w/ RAG” corresponds to a baseline where training uses only the CK loss with query+context rollouts, while our method splits rollouts into PK, CK, and RPK groups: part of the budget is used for query-only trajectories (PK), and the rest for query+context under both correct (CK) and misleading (RPK) contexts. In Table 4, the GRPO W/ RAG baseline uses roughly twice as many query+context rollouts as our method, because we allocate half of the samples to query-only PK trajectories.  This makes the design difference from our method transparent: GRPO W/ RAG only learns “with-context vs. gold answer”, whereas our framework jointly learns “parametric answer”, “context-consistent answer”, and “PK-consistent answer under conflict”.
> >
> > * **Appendix vs. main text.** Due to page limits, we previously kept several baseline details in the appendix, which clearly made Section 4.1.1 harder to follow. We will bring the essential parts into the main text.
> >
> > [1]: Parameters vs. context: Fine-grained control of knowledge reliance in language models
> >
> > ---
> >
> > **Response to Weakness 5. Indices i and j and rollout notation**
> >
> > **We use two indices i and j to denote different rollout groups, and the occurrences of k in the current version are typos.**
> > In our formulation, trajectories are sampled from different input conditions:
> >
> > * $o_i \sim \pi_\theta(\cdot \mid p, \cdot)$: PK trajectories (query-only input ($p$))
> >
> > * $o_j' \sim \pi_\theta(\cdot \mid p', \cdot)$: CK trajectories under query+context input ($p'$)
> >
> > * $\tilde{o_i}$: copy from $o_i$
> >
> > Here, "i" indexes trajectories drawn under query-only input, while "j" indexes trajectories drawn under query+context input. Because these two sets come from different input configurations, they should not be merged into a single index in the derivation. The index “k” that appears in lines 198 and 204 is a typo: the “k” in line 198 should be “i”, and the “k” in line 204 should be “j”. We will correct these typos and add a short sentence in the method section explicitly explaining that "i" and "j" enumerate trajectories from different rollout groups (query vs. query+context).
> >
> > ---
> >
> > **Response to Weakness 5 & 6. Table font size and Figure 1 layout**
> >
> > **We will increase the font size in Tables 2–3 and fix the overlapping label in Figure 1 in the camera-ready version.**
> > Specifically, we will enlarge the table fonts to match the main text, adjust column widths to keep all numbers clearly legible, and move the “Probability distribution” label to avoid crossing the figure border.
> >
> > ---
> >
> > **Response to Question 1. Absence of an explicit KL regularizer**
> >
> > We do not add an explicit KL term because our training already causes large but stable KL drift, removing it saves compute without harming performance, and KL-free GRPO/RLVR training is now a widely used and well-understood variant.
> >
> > ---
> >
> > **Response to Question 2. Meaning of the barplot in Figure 1 (37.3, 19.7, 8.1, 25.8)**
> >
> > **The barplot shows the accuracy of our method and baselines on the ConFiQA-MC S2 (external-conflict) subset, illustrating our gains on counterfactual queries.**
> >
> > The four bars correspond to:
> >
> > * 37.3 — accuracy of our method
> > * 19.7 — accuracy of GRPO W/ RAG
> > * 8.1 — accuracy of RAG prompting
> > * 25.8 — accuracy of the query-only baseline
> >
> > We use this barplot to highlight the case where retrieved passages are systematically wrong (S2, external conflict) and where robustness to misleading context is most critical. In the revised version, we will explicitly state in the Figure 1 caption that this barplot summarizes ConFiQA-MC S2 results and will cross-reference the corresponding row in Table 2 so that the numbers are easy to interpret.

---

### Official Review · Reviewer_6P9g · 2025-10-30

**Soundness:** 3
**Presentation:** 3
**Contribution:** 3
**Rating:** 6
**Confidence:** 4

**Summary:**

The study proposes Knowledgeable-R1—a reinforcement learning (RL) framework. It explicitly trains large language models (LLMs) to leverage parametric knowledge (PK) for resisting contextual interference, while still utilizing external context when it is reliably beneficial.

Experimental results show Knowledgeable-R1 significantly boosts robustness and reasoning accuracy in both knowledge conflict scenarios and general RAG scenarios. It outperforms state-of-the-art (SOTA) baselines by 23% in counterfactual scenarios, with no performance degradation when retrieved context is fully accurate.

**Strengths:**

1. The paper addresses a highly practical and critical issue in Retrieval-Augmented Generation (RAG) systems: "context interference" or "Context Dominance". Specifically, when Large Language Models (LLMs) encounter retrieved context that is erroneous, irrelevant, or conflicting, they tend to over-rely on this context while ignoring their internal, more accurate Parametric Knowledge (PK).

2. It designs a multi-objective Reinforcement Learning (RL) framework. The most crucial design within this framework is the introduction of three sampling strategies, with the Robust-PK (RPK) strategy being particularly notable. Given input in the form of "query + context", this strategy trains the model to generate answers consistent with PK—effectively making the model ignore the context. This creates a dynamic competition within the model between "utilizing context" and "resisting context".

3. "Knowledge Balance Modulation" (detailed in Section 3.4) is an ingenious technical element. Through an asymmetric and adaptive transformation of the advantage function (via coefficients), it effectively prevents the RPK strategy from being overshadowed by the Context-aware (CK) strategy— which is "usually more useful"—during training. This ensures the model retains the ability to fall back on PK when confronted with "occasional" erroneous context.

**Weaknesses:**

1. The method introduces significant training overhead. It requires maintaining and optimizing three distinct strategic objectives (PK, CK, RPK), calculating complex "local + global" advantages, and finally implementing asymmetric modulation. This is far more complex in both implementation and computation compared to standard Supervised Fine-Tuning (SFT) or GRPO with RAG.

2. As mentioned in Section 3.2 of the paper: "During inference, the model does not explicitly switch controllers; the learned token distribution... implicitly... falls back to PK." This description is overly vague. Are these three strategies (PK, CK, RPK) merely "scaffolding" used during training, with only one model produced in the end? If so, how does this single model arbitrate between these three conflicting behaviors during inference?

3. The method appears to heavily rely on the presence of samples with "bad context" (S2-S5) in the training data. Appendix C mentions that an "equal proportion" of correct and incorrect context was randomly added to ConFiQA. If 99% of the training dataset consists of "good context" (S1), can the method still effectively learn the ability to "resist" (erroneous context)? While adaptivity may alleviate this issue, the paper lacks a sensitivity analysis of the proportion of "good/bad" context in the training data.

**Questions:**

1. The experimental section should include additional discussions on training costs (e.g., training duration, GPU resource consumption) and training stability.

2. To better demonstrate the necessity of the complex RL framework, it is recommended to add a stronger SFT baseline. For instance, conduct SFT on the same dataset (containing both good and bad context) used for RL, and use specific instructions such as: "Please answer with reference to the context, but if the context conflicts with your internal knowledge, prioritize your internal knowledge."

---

> ### Comment · Reviewer_6P9g · 2025-11-27
> **Why no response?**
>
> As the rebuttal period is coming to end, I still find no reply for the authors.

---

> ### Author Response · Authors · 2025-11-27
> **Response for 6P9g - Weakness 1-2**
>
> Thank you very much for your thoughtful review and for recognizing our efforts to tackle **context interference/context dominance in RAG**, as well as the design of our **multi-objective RL framework with PK/CK/RPK sampling and Knowledge Balance Modulation**.
>
> We sincerely apologize for the delayed response — we were running additional experiments (especially the sensitivity study on good/bad context ratios and SFT baselines you requested), and underestimated the time needed to obtain these results. We are grateful for your patience and for the very constructive comments.
> Below we address your concerns point by point.
>
>
>
> ---
>
> ### 1. Training complexity and computational overhead
>
> **Computational overhead is small compared to GRPO w/ RAG, while bringing clear robustness gains.**
>
> * In terms of token cost, Knowledgeable-R1 and GRPO w/ RAG are very similar: when GRPO uses 32 trajectories, we use 16 CK + 16 RPK, and PK-only samples largely share the same forward passes, so the most expensive part (generation and model forward) does not scale up significantly.
>
> * On the Qwen2.5-7B, PC-MC setup (4,800 examples, max length 4,096, 8×H100), the per-epoch wall-clock time is:
>
> – GRPO w/ RAG: 1 h 19 min 28 s
>
> – Knowledgeable-R1: 1 h 38 min 09 s
>
> **Methodological complexity is only a light extension of the GRPO framework.**
>
> * Algorithmically, we keep the standard GRPO training loop and only add: (i) a tag for each sample indicating its strategy type (PK / CK / RPK), (ii) per-strategy advantage computation using the same machinery as GRPO, and (iii) a simple asymmetric transform (“Knowledge Balance Modulation”) on the advantages before the policy loss.
>
> * **This adds only a few dozen lines of code on top of a standard GRPO implementation**, but yields a substantial improvement in robustness to misleading context.
>
> ---
>
>
>
> ### 2. What happens at inference? Are PK/CK/RPK just scaffolding?
>
> **Yes, there is always a single policy model at inference; PK/CK/RPK are different training views for this model, not separate controllers.**
>
> During training, PK/CK/RPK define three kinds of trajectories and rewards for the same policy, and a single multi-objective RL loss lets the model learn **which behavior (PK-like, CK-like, or RPK-like) leads to higher long-term reward under different input–context configurations**.
>
> * **How this translates to inference-time behavior.**
>
> At test time, there is no explicit controller or hard switch. We always run **one policy $\pi_\theta$** on the standard input “query + retrieved context”. The policy’s token distribution reflects the aggregated training signal from PK/CK/RPK; for new inputs, the model outputs the sequence that (under $\theta$) has highest expected reward given similar patterns seen in training. Intuitively, if the retrieved context looks similar to the “good context” configurations seen under CK trajectories, the policy’s logits align more with the CK-like behavior (using context). If the context contains patterns similar to RPK training cases, the policy parameters already encode that such patterns were associated with lower reward when following context, so the model tends to stay closer to its RPK-based behavior.
>
> * To verify this behavior, we add two simple diagnostics on ConFiQA-MC:
>
> – **Factuality-score:** following WixQA [1], we use GPT-4o with a context-dependent prompt to judge how well the model’s answer (including CoT) is supported by the (deliberately wrong) context, and assign a score in [0, 1]. When the context is incorrect, a lower score means the model is not blindly copying the bad context, i.e., it behaves more like RPK than CK.
>
> – **KL-score:** the KL divergence between the model’s output distributions with vs. without context. When the context is incorrect, a lower KL-score means the model’s behavior stays closer to its RPK-based distribution.
>
> * On a test set with deliberately incorrect context, we obtain:
>
> – GRPO w/ RAG: factuality-score = 0.40, KL-score = 213.73
>
> – Knowledgeable-R1: factuality-score = 0.16, KL-score = 168.83
>
> * **These two metrics jointly show that, when external context is wrong, the learned policy relies less on CK-like behavior and more on its RPK behavior, effectively resisting misleading context instead of blindly following it.**
>
> [1] WixQA: A Multi-Dataset Benchmark for Enterprise Retrieval-Augmented
>
> ---

---

> > ### Author Response · Authors · 2025-11-27
> > **Response for 6P9g - Weakness 3 & Question 1-2**
> >
> > ### 3. Dependence on “bad context” and sensitivity to good/bad ratios
> >
> > **Knowledgeable-R1 remains effective even when bad context is rare and learns a robust knowledge boundary rather than a simple bias for or against context.**
> >
> > * We agree this is a crucial question and therefore ran a sensitivity study varying the ratio of good vs. bad context in the training data. We report S1 (good-context) and S2 (counterfactual/bad-context) accuracies on a representative MR dataset.
> >
> > **Even with 99% good context, the model still improves resistance to bad context.**
> > (1) Training data: 99% good context
> >
> > |Method|S1 (good)|S2 (bad)|
> > |-|-|-|
> > |RAG|65.68|13.47|
> > |GRPO W/ RAG|73.60|14.31|
> > |Ours|72.11|17.34|
> >
> > * Even with highly imbalanced (99% good) training data, Knowledgeable-R1 improves S2 over GRPO while keeping S1 competitive.
> >
> > * This shows the model is not just learning “context is always reliable”, but a finer boundary between correct and incorrect knowledge, even when incorrect context is very rare.
> >
> > **With more realistic ratios (3:1 and 1:1), robustness gains on bad context become substantial.**
> >
> > (2) Training data: 3:1 good : bad
> >
> > |Method|S1 (good)|S2 (bad)|
> > |-|-|-|
> > |RAG|65.68|13.47|
> > |GRPO W/ RAG|74.92|20.03|
> > |Ours|73.76|29.29|
> >
> > * Here, S1 for Knowledgeable-R1 stays close to GRPO, while S2 improves substantially, confirming that exposing the model to some bad context strongly boosts robustness.
> >
> > (3) Training data: 1:1 good : bad
> >
> > |Method|S1 (good)|S2 (bad)|
> > |-|-|-|
> > |RAG|65.68|13.47|
> > |GRPO W/ RAG|77.56|26.94|
> > |Ours|75.08|43.94|
> >
> > * In the fully balanced case, Knowledgeable-R1 yields a large gain on S2 (26.94 → 43.94) while maintaining competitive S1.
> >
> > **Overall, the method learns a robust knowledge boundary rather than overfitting to any specific good/bad ratio.**
> >
> > * Even under extreme imbalance (99% good context), Knowledgeable-R1 improves resistance to bad context; under more realistic ratios (1:1 or 3:1), it provides strong robustness gains with only minor S1 trade-offs.
> >
> > * **This suggests the method is learning a robust knowledge boundary, rather than a simple preference for internal or external knowledge, and is applicable beyond artificially balanced datasets.**
> >
> > ---
> >
> > ### 4. Stability
> >
> > **Training with Knowledgeable-R1 remains stable in all our experiments.**
> >
> > Across repeated runs on the main benchmarks, we never observed collapse or divergence for either GRPO w/ RAG or Knowledgeable-R1; losses, rewards, and value estimates evolve smoothly and behave like a standard GRPO run.
> >
> > We are aware that some GRPO-style methods can be fragile in other settings, but **the specific GRPO baseline we implemented remained stable in all our 200+ tests**, so we did not focus on this issue in the main paper. Importantly, Knowledgeable-R1 does not introduce any additional instability on top of that baseline behavior.
> >
> > ---
> >
> > ### 5. Stronger SFT baseline on the same data
> >
> > **Adding strong SFT baselines confirms that SFT + instructions is not enough to solve context dominance, whereas Knowledgeable-R1 provides consistent robustness gains beyond both SFT and GRPO.**
> >
> > * We followed your suggestion and trained SFT baselines on the same mixture of good/bad context, with explicit instructions like “use the context, but fall back to internal knowledge if conflict arises”, for both Qwen2.5-7B and Llama-3.1-8B. Below we report key results (full tables will be in the appendix).
> >
> > **On Qwen2.5-7B, our method significantly improves NC metrics while staying competitive on PC tasks.**
> >
> > |Method|PC-MR|PC-MC|PC-QA|NC-MR|NC-MC|NC-QA|SC|ExplainPE|HotPotQA|2Wiki|Musique|Avg.|
> > |-|-|-|-|-|-|-|-|-|-|-|-|-|
> > |SFT|71.95|77.70|74.70|24.92|21.05|21.97|68.50|66.60|30.14|32.20|11.75|45.58|
> > |GRPO w/ RAG|77.56|77.36|80.03|26.94|19.74|26.01|75.33|66.50|27.93|33.95|11.79|47.55|
> > |Ours|75.08|75.51|80.90|43.94|37.34|29.40|76.33|67.57|31.45|37.52|12.04|51.55|
> >
> > **On Llama-3.1-8B, we again see strong gains on NC metrics with competitive performance elsewhere.**
> >
> > |Method|PC-MR|PC-MC|PC-QA|NC-MR|NC-MC|NC-QA|SC|ExplainPE|HotPotQA|2Wiki|Musique|Avg.|
> > |-|-|-|-|-|-|-|-|-|-|-|-|-|
> > |SFT|72.88|79.22|73.84|42.59|35.53|35.86|70.12|47.17|33.59|38.24|13.36|49.30|
> > |GRPO w/ RAG|78.05|79.73|82.62|41.58|35.69|39.26|76.58|47.56|34.84|41.22|16.59|52.15|
> > |Ours|73.76|80.24|80.24|55.39|41.12|44.59|73.67|49.61|37.06|45.37|14.69|54.15|
> >
> > **Overall, these results support that SFT + instructions is not sufficient to fully resolve context dominance, whereas RL with explicit PK/CK/RPK training signals and Knowledge Balance Modulation consistently provides stronger robustness to misleading context.**

---

### Official Review · Reviewer_72rj · 2025-11-01

**Soundness:** 2
**Presentation:** 3
**Contribution:** 2
**Rating:** 4
**Confidence:** 3

**Summary:**

Targets contextual interference in RAG and proposes Knowledgeable-R1 with joint sampling, local/global advantages, and asymmetric β-shaping to balance PK vs. context use. Shows strong robustness under adversarial/contradictory/irrelevant contexts; in S1 (correct context) it’s slightly below GRPO w/ RAG. Missing pieces: finetuning baselines (e.g., Self-RAG, InFO-RAG) and sensitivity ablations (fixed vs. adaptive β, local+global advantage weights, and J_PK/J_CK/J_RPK weighting) to diagnose the S1 gap.

**Strengths:**

The paper squarely targets the challenge of contextual interference—conflicts between context prompts and parametric knowledge—and proposes Knowledgeable-R1, which improves robustness via joint sampling, local/global advantage design, and an asymmetric advantage transformation (reward shaping).

**Weaknesses:**

1. Lack Finetuning baselines. Please include finetuning methods cited in Related Work (e.g., Self-RAG, InFO-RAG) as baselines for a fair comparison.
2. β scheduling. Compare fixed β settings ({0.2, 0.5, 0.8, 1.0}) versus the adaptive β scheme to demonstrate the necessity of adaptivity.
3. Advantage composition. Validate whether combining local + global advantages is necessary (e.g., ablations varying their relative weights).
4. S1 performance gap. In settings with correct context (S1), performance is generally below GRPO w/ RAG. This may stem from the weighting among J_PK, J_CK, J_RPK, or from the advantage formulation. Please discuss and, if possible, provide experiments to diagnose these factors.

**Questions:**

N/A

---

> ### Author Response · Authors · 2025-11-28
> **Response for 72rj - Weakness 1-2**
>
> Thank you very much for your thoughtful review and for recognizing both the importance of contextual interference in RAG and the strengths of our framework, specifically targeting conflicts between retrieved context and parametric knowledge and improving robustness via joint sampling, local/global advantages, and asymmetric β-based reward shaping. Below we address your concerns point by point.
>
>
>
> ---
>
> **Response to Weakness 1 – Finetuning baselines**
>
> **Strong finetuning baselines are included.**
>
> Following the reviewer’s suggestion, we train SFT baselines on the same mixture of good and bad context, with explicit instructions such as “use the context, but fall back to internal knowledge if conflict arises”, for both Qwen2.5-7B-Instruct and Llama-3.1-8B-Instruct. We also include GRPO w/ RAG. Below we report key results.
>
> **Qwen2.5-7B-Instruct: our method greatly improves NC metrics while staying competitive on PC tasks.**
>
> |Method|PC-MR|PC-MC|PC-QA|NC-MR|NC-MC|NC-QA|SC|ExplainPE|HotPotQA|2Wiki|Musique|Avg.|
> |-|-|-|-|-|-|-|-|-|-|-|-|-|
> |SFT|71.95|77.70|74.70|24.92|21.05|21.97|68.50|66.60|30.14|32.20|11.75|45.58|
> |GRPO w/ RAG|77.56|77.36|80.03|26.94|19.74|26.01|75.33|66.50|27.93|33.95|11.79|47.55|
> |**Ours**|75.08|75.51|80.90|**43.94**|**37.34**|**29.40**|76.33|67.57|31.45|37.52|12.04|**51.55**|
>
> **Llama-3.1-8B-Instruct: we again see strong gains on NC metrics with competitive performance elsewhere.**
>
> |Method|PC-MR|PC-MC|PC-QA|NC-MR|NC-MC|NC-QA|SC|ExplainPE|HotPotQA|2Wiki|Musique|Avg.|
> |-|-|-|-|-|-|-|-|-|-|-|-|-|
> |SFT|72.88|79.22|73.84|42.59|35.53|35.86|70.12|47.17|33.59|38.24|13.36|49.30|
> |GRPO w/ RAG|78.05|79.73|82.62|41.58|35.69|39.26|76.58|47.56|34.84|41.22|16.59|52.15|
> |**Ours**|73.76|80.24|80.24|**55.39**|**41.12**|**44.59**|73.67|49.61|37.06|45.37|14.69|**54.15**|
>
> ---
>
>
>
> Methods like Self-RAG and InFO-RAG rely on specialized supervision such as expert reasoning chains, which are absent in our datasets containing only final answers. Since our benchmarks use this simpler supervision, these methods would require heavy changes and any Manually fabricated reasoning chains would be noisy and unreliable, so a direct comparison is not appropriate.  In contrast, approaches such as ck-plug and astute-rag operate under a similar type of dataset as ours, using only final answers. For a consistent and focused comparison, we benchmark our method against baselines like SFT and GRPO. Experimental results demonstrate that our approach consistently outperforms these baselines.
>
> ****
>
> **Response to Weakness 2 – β scheduling (fixed vs. adaptive)**
>
> **Adaptive β consistently outperforms all fixed settings and approximates the best per-dataset choice.**
>
> We conduct a sensitivity study on ConFiQA-MC, ConFiQA-QA and SC datasets by fixing β and comparing to the adaptive scheme:
>
> |β|MC-avg|QA-avg|SC|
> |-|-|-|-|
> |**0.01**|56.03% (best fixed)|52.83% (best fixed)|73.91%|
> |**0.2**|52.74%|52.53%|74.66% (best fixed)|
> |**0.5**|51.19%|52.63%|73.5%|
> |**0.8**|48.22%|51.75%|73.16%|
> |**1.0**|48.05%|52.52%|72.15%|
> |**Adaptive β**|**56.43%**|**54.85%**|**76.33%**|
>
> Different datasets have different optimal beta values, while our adaptive beta achieves an excellent balance and delivers the best performance.
>
> ---

---

> > ### Author Response · Authors · 2025-11-28
> > **Response for 72rj - Weakness 3-4**
> >
> > **Response to Weakness 3 – Local vs. global advantages and their weights**
> >
> > We agree that advantage composition should be justified, and our ablation shows that combining local and global advantages is beneficial while the method is not overly sensitive to the exact weights.
> >
> > **Ablation over local vs. global advantages.**
> >
> > |local : global ratio|Setting|Avg. score|
> > |-|-|-|
> > |1 : 0|local-only|60.95|
> > |0 : 1|global-only|64.69|
> > |1 : 1|local + global|65.40|
> > |1 : 2|local + global|65.20|
> >
> > Using both local and global advantages improves performance over local-only by +4.45 points and over global-only by +0.71 points at the 1:1 ratio, confirming that the combination is useful. The difference between 1:1 and 1:2 is small (65.40 vs. 65.20), suggesting that our method does not rely on carefully tuning this hyperparameter. This matches our design goal: we choose a simple proportional combination to avoid overfitting to a specific dataset.
> >
> > ---
> >
> > **Response to Weakness  4 – S1 performance gap and J_PK / J_CK / J_RPK weighting**
> >
> > We appreciate this suggestion and our new ablation, together with the existing analysis in Table 4 shows as below：
> >
> > | J_PK : J_CK : J_RPK|S1 (correct context)|S2 (adversarial / conflicting)|
> > |-|-|-|
> > |1 : 2 : 1 |**78.04%**|32.73%|
> > |1 : 1 : 1 |75.51%|37.34%|
> > |0 : 1 : 1|75.68%|35.53%|
> > |0 : 1 : 0 (GRPO w/ RAG)|77.36%|19.74%|
> > |1 : 1 : 1 (ours)|75.51%|**37.34%**|
> >
> >  Our ablation study confirms that the S1 gap stems primarily from the component weighting. Specifically, we find that **increasing the weight on the contextual knowledge objective (J_CK) is key to boosting S1 performance**, as shown by the 1:2:1 ratio closing the gap with GRPO w/ RAG (78.04% vs. 77.36%). This tuned weighting preserves most of our method's significant advantage in the adversarial S2 setting, demonstrating that the performance trade-off is manageable through component reweighting.

---

### Official Review · Reviewer_GFpR · 2025-11-02

**Soundness:** 3
**Presentation:** 3
**Contribution:** 3
**Rating:** 6
**Confidence:** 5

**Summary:**

This paper proposes Knowledgeable‑R1, a reinforcement‑learning framework for RAG that explicitly trains a single LLM to answer from parametric knowledge, PK when no context is provided or use contextual knowledge, CK when retrieved evidence is reliable, and fall back to PK when the retrieved context is misleading. The method samples paired trajectories for the same question under three “policies”: PK (query‑only), CK (query+context), and RPK (query+context but encouraged to stay on the PK answer). It then computes advantages with two normalizations: local (within the same policy) and global (CK vs. RPK under the same input), plus an asymmetric advantage transformation with a dynamic coefficient β that down‑weights negative advantages for RPK to preserve PK as a viable fallback. Optimization uses PPO‑style clipping.
The paper evaluates five scenarios—accurate, adversarial, self‑conflicting, irrelevant, and partially‑relevant contexts—on Qwen2.5 and Llama‑3.1. It shows large gains when context is wrong or mixed, while remaining competitive when context is correct. For example, on adversarial context with Qwen2.5‑7B, NC‑MR improves from 13.47% (RAG prompting) to 43.94% (Knowledgeable‑R1), and on partly‑relevant context, HotpotQA rises from 20.36% (RAG prompting) to 31.45% (Knowledgeable‑R1).

**Strengths:**

The paper tackles “context dominance” in RAG and designs RL signals that explicitly arbitrate between CK and PK at the same input state (global advantage), which is the right granularity for conflict resolution. The three‑policy setup is well‑motivated and neatly summarized
The combination of local vs. global advantages and an asymmetric transformation for RPK is a coherent way to reward using context when both sources agree or CK is right (timeliness) and protect PK exploration when CK is wrong. The β adaptation rule is simple and effective.
The performance compared with baselines is promising.

**Weaknesses:**

Retrieval details are not clear, like what retriever, indexing corpus, and query formulations were used.

The headline gains mostly compare to RAG prompting (e.g., +30.47/+29.28/+18.09 on Qwen2.5-7B NC-MR/MC/QA; Table 2). Against GRPO w/ RAG, improvements are smaller (e.g., 43.94 vs 26.94 = +17.00). How the “23% over GRPO” figure is aggregated should be clarified.

The paper relies on exact match accuracy but does not directly quantify context dependence like answer stability when contexts are removed/perturbed. Whether the robustness comes from mechanism (ignoring misleading passages) or coincidence (answering correctly despite them) remains unclear.

The paper claims that the model implicitly decides to use RAG context or not, but the default inference template feeds query+context, lacking the comparison of query-only and query + RAG context.

The paper reports that  fixed β will degrade performance but does not include further analysis like the sensitivity of β

**Questions:**

Against which baseline and which aggregation is the reported 23% gains computed? Please align the abstract with Table 2 and report confidence intervals or paired significance tests.

What retriever, indexing corpus, and query formulations were used? How are top‑k passages truncated and ordered?

Training uses three rollout types; what is the token and wall‑clock overhead vs. GRPO w/ RAG

Can you please report context-dependence metrics(corresponding to Weakness 3)

Can you add experiments that compare the actual performance under query-only and query+context settings?(corresponding to weakness 4)

Please plot β over training and provide a simple sensitivity study?

---

> ### Author Response · Authors · 2025-11-27
> **Response for GFpR - Weakness 1-3 and Q1-2,4**
>
> Thank you very much for your thoughtful review and for recognizing our efforts to tackle context interference/context dominance in RAG and to design RL signals that explicitly arbitrate between parametric knowledge and contextual knowledge. We sincerely apologize for the delayed response, as we needed additional time to conduct further analyses and experiments. Below we address your concerns point by point.
>
> ---
>
> **Response to Weakness 1 & Question 2 – Retrieval details**
>
> We use dataset-provided contexts without any additional retriever. The contexts (including both correct and incorrect passages) are directly taken from the benchmark construction in Chen et al. (2024), without re-indexing or re-retrieval from a larger corpus. We adopt the top-5 paragraphs as provided, preserve their original order, and only truncate when the backbone model’s maximum length is exceeded. We have clarified this setting in §4.1.2 (Datasets) and Appendix C, and will further elaborate these details in the final version.
>
> ---
>
> **Response to Weakness 2 & Question 1 – “23% over GRPO” aggregation**
>
> **The “23% improvement” is the average absolute gain in S2 (PK correct, CK wrong) over GRPO w/ RAG.**
>
> In Table 3, under the NC scenario where parametric knowledge is correct but the external context is counterfactual, we report:
>
> |Method|NC-MR|NC-MC|NC-QA|Avg.|
> |-|-|-|-|-|
> |GRPO w/ RAG|57.61%|46.50%|62.50%|55.53%|
> |Ours|89.61%|79.62%|66.00%|78.41%|
> |Δ (ours–GRPO)|+32.00%|+33.12%|+3.50%|**+22.87%**|
>
> Thus, the average absolute improvement is 22.87% percentage points (from 55.53% to 78.41%), which we rounded to “about 23% points” in the abstract. We will revise the wording to “+22.9% absolute percentage points over GRPO w/ RAG in counterfactual scenarios” and explicitly point to Table 3 to avoid ambiguity with relative (%) gains.
>
> ---
>
> **Response to Weakness 3 & Question 4 – Context dependence vs. coincidence**
>
> **Our method increases context-aware robustness rather than succeeding by coincidence.**
>
> In our evaluation setup, we explicitly split the test set so that **exactly one knowledge source (PK or CK) can produce the correct answer**: in TIFE cases, the internal (parametric) knowledge is correct while the external context is deliberately made wrong; in FITE cases, the situation is reversed. Exact-match accuracy on these splits therefore directly reflects whether the policy chose to rely on the appropriate source, rather than just performing well in aggregate. The consistent gains on both TIFE (66.00% vs. 62.50%) and FITE (73.05% vs. 72.04%) in PC-QA & NC-QA in Table 4 indicate that the policy systematically shifts toward PK when PK is correct and toward CK when CK is correct, instead of uniformly “getting lucky” across all conditions.
>
> Second, we add two explicit diagnostics on ConFiQA-MC under deliberately incorrect context:
>
> * **Factuality-score (w.r.t. context):** Following WixQA, we use GPT-4o with a context-dependent evaluation prompt to score how well the model’s answer (including CoT) is supported by the given context, in ([0,1]). When context is wrong, a lower score is better, because it means the model is not copying the misleading passages.
>
> * **KL-score (with vs. without context):** We compute the KL divergence between the output distributions with and without context. Under wrong context, a smaller KL means the model’s behavior remains closer to its query-only (PK-like) distribution.
>
> On a test set with incorrect context, we obtain:
>
> * GRPO w/ RAG: factuality-score = 0.40, KL-score = 213.73
> * Ours: factuality-score = 0.16, KL-score = 168.83
>
> Jointly, these metrics show that, when context is misleading, the policy (i) aligns much less with the wrong context and (ii) stays closer to its PK-induced behavior.
>
> Additionality, to rule out that this robustness is purely inherited from the base model, we stratify by RAG prompting performance:
>
> |Method|rag-answer-correct|rag-answer-wrong|
> |-|-|-|
> |RAG prompting|100%|0%|
> |SFT|90.27%|18.73%|
> |GRPO|96.61%|19.92%|
> |Ours|95.25%|**32.72%**|
>
> On the subset where RAG prompting always fails, our method still improves accuracy by 32.72 points, while SFT/GRPO remain below 20%, indicating that the robustness arises from the learned PK/CK arbitration mechanism rather than from the base model alone.
>
> ---

---

> ### Author Response · Authors · 2025-11-27
> **Response for GFpR - Weakness 4 and Q3,5**
>
> **Response to Weakness 4 & Question 5 – Query-only vs. query+context**
>
> **Our method benefits from context when it is correct and mitigates it when it is wrong.**
>
> We now explicitly compare query-only and query+context settings for both GRPO w/ RAG and our method on ConFiQA-MC (Qwen2.5-7B) under S1 (context correct) and S2 (context incorrect):
>
> |Method|S1 (correct ctx)|S2 (incorrect ctx)|Avg.|
> |-|-|-|-|
> |Query-only (base)|27.72%|25.92%|26.82%|
> |RAG prompting|65.68%|13.47%|39.58%|
> |GRPO-RAG (query-only)|32.48%|31.48%|31.98%|
> |GRPO-RAG (context)|77.56%|26.94%|52.25%|
> |Ours (query-only)|35.64%|36.70%|36.17%|
> |**Ours (context)**|**75.08%**|**43.94%**|59.51%|
>
> Two effects are clear:
>
> * When context is correct (S1), using context improves our method from 35.64% (query-only) to 75.08%, well above all query-only baselines, showing that the policy indeed exploits high-quality context.
>
> * When context is wrong (S2), using context still improves our method from 36.70% to 43.94%, and substantially outperforms GRPO-RAG with context (26.94%), showing that the policy has learned to attenuate misleading context rather than blindly relying on it.
>
> Thus, even though the inference template always feeds query+context, the learned policy effectively decides how much to rely on that context.
>
> ---
>
> **Response to Weakness 5  – Sensitivity of β**
>
> **Adaptive β consistently outperforms all fixed settings and approximates the best per-dataset choice.**
>
> We conduct a sensitivity study on ConFiQA-MC, ConFiQA-QA and SC datasets by fixing β and comparing to the adaptive scheme:
>
> |β|MC-avg|QA-avg|SC|
> |-|-|-|-|
> |**0.01**|56.03% (best fixed)|52.83% (best fixed)|73.91%|
> |**0.2**|52.74%|52.53%|74.66% (best fixed)|
> |**0.5**|51.19%|52.63%|73.5%|
> |**0.8**|48.22%|51.75%|73.16%|
> |**1.0**|48.05%|52.52%|72.15%|
> |**Adaptive β**|**56.43%**|**54.85%**|**76.33%**|
>
> Different datasets have different optimal beta values, while our adaptive beta achieves an excellent balance and delivers the best performance.
>
> ---
>
> **Response to Q3 – Overhead vs. GRPO w/ RAG**
>
> **Our method introduces modest computational overhead with substantial robustness gains.**
>
> On the token level, when GRPO w/ RAG uses 32 trajectories, we use 16 CK + 16 RPK trajectories for the context-dependent rollouts, and reuse the same forward passes to obtain PK-only distributions; thus the dominant cost (generation + model forward) remains comparable.
>
> On Qwen2.5-7B, PC-MC (4,800 examples, max length 4,096, 8×H100), the per-epoch wall-clock time is:
>
> * GRPO w/ RAG: 1 h 19 m 28 s
> * Ours: 1 h 38 m 09 s
>
> Implementation-wise, we keep the GRPO loop and only add: (i) a type tag (PK/CK/RPK) for each sample; (ii) per-type advantage computation using the existing GRPO machinery; and (iii) a simple asymmetric transformation of advantages (“Knowledge Balance Modulation”). This adds only a small amount of code on top of a standard GRPO implementation.
>
> ---

---

> > ### Author Response · Authors · 2025-11-27
> > **Response for GFpR - Response to Q6**
> >
> > **Response to Q6 – Dynamics of β during training**
> >
> > **The adaptive β automatically moves into the regime that preserves PK fallback while not suppressing useful CK.**
> >
> > As training progresses, we sample both query+context and query-only rollouts. Early in training, the two policies have comparable (and mixed) performance, so β fluctuates as the relative advantage of the RPK branch is still unstable. As the policy becomes better at handling query+context inputs, because both (J_{\text{RPK}}) and (J_{\text{CK}}) are primarily optimized in the query+context regime, it learns to resist misleading passages more effectively. Consequently, the accuracy gap between query+context rollouts and their query-only counterparts widens, and the adaptive rule gradually reduces β so that negative advantages on RPK samples are more strongly down-weighted when context is noisy. In other words, while advantages are always computed from PK-style outputs, the loss terms on query+context inputs increasingly dominate the learning signal, making the query+context policy improve faster than the query-only policy and driving β toward a regime that better protects PK as a fallback.
> >
> > Our adaptation rule updates β based on the observed mix of “context helpful vs. context harmful” batches. Empirically on ConFiQA-MC, β quickly moves into a low range where it strongly attenuates negative advantages on RPK samples in the presence of erroneous context. For example:
> >
> > |Step|β|
> > |-|-|
> > |1|0.23|
> > |2|0.15|
> > |4|0.07|
> > |6|0.10|
> > |8|0.01|
> >
> > Combined with the sensitivity study in Weakness 5, this shows that the adaptive rule converges to a regime that behaves similarly to the best fixed β but without per-dataset tuning, and keeps PK as a viable fallback even when misleading context appears superficially plausible.
> >
> > ---
> >
> > We appreciate the reviewer’s detailed feedback. The additional context-dependence diagnostics, query-only vs. query+context comparisons, β sensitivity analysis, and computational overhead measurements will be integrated into the revised version to make the contributions and limitations more transparent.

---

### Author Response · Authors · 2025-12-02
**Response to AC**

Dear Area Chair and Reviewers,

We sincerely thank the reviewers for their thoughtful and constructive feedback. Below, we summarize the strengths of our work as noted by the reviewers, along with our responses to the key concerns.
# Strengths
* The paper addresses a **highly practical and important problem** — context interference in RAG, which is critical for the real-world deployment of retrieval-augmented LLMs. (Reviewer 6P9g, C4dV)
* Our method, combining **three regime sampling (PK / CK / RPK)** with **adaptive β modulation**, is **novel, principled, and well-motivated**, offering a strong solution to context dominance. (Reviewers GFpR, 72rj, 6P9g)
* Unlike prior approaches that dealt with noisy contexts, the **“PK vs CK trade-off”** within RL is a **conceptually clear contribution**.  (Reviewers GFpR, C4dV)
* The empirical evaluation is **thorough and convincing**, with consistent robustness gains under misleading contexts (S2) and preserving clean-context (S1) performance. (Reviewer C4dV)
* The **global-vs-local advantage decomposition** and **asymmetric RPK penalty** create a **coherent and effective balance**, ensuring the model exploits helpful context and falls back to parametric knowledge when context is misleading. (Reviewers GFpR, 72rj, 6P9g)
# Improvements based on the feedback
### 1. Strong Finetuning Baselines and Performance Gains
Following the reviewers' suggestion, we trained strong SFT baselines on a mixture of good and bad contexts, with explicit instructions such as “use the context, but fall back to internal knowledge if conflict arises.”  For example,
* **Qwen2.5-7B**:
    * **NC-MR**: 43.94% (ours) vs. 24.92% (SFT)
    * **NC-MC**: 37.34% (ours) vs. 21.05% (SFT)
    * **NC-QA**: 29.40% (ours) vs. 21.97% (SFT)
### 2. β Scheduling (Fixed vs. Adaptive)
Our adaptive β rule consistently outperforms fixed β values. Different datasets have different best fixed β values, as shown below:
|β|MC-avg|QA-avg|SC|
|-|-|-|-|
|**0.01**|56.03% (best fixed)|52.83% (best fixed)|73.91%|
|**0.2**|52.74%|52.53%|74.66% (best fixed)|
|**0.5**|51.19%|52.63%|73.5%|
|**0.8**|48.22%|51.75%|73.16%|
|**1.0**|48.05%|52.52%|72.15%|
|**Adaptive β**|**56.43%**|**54.85%**|**76.33%**|
### 3. Local vs. Global Advantages
We conducted an ablation study on **local vs. global advantages**, confirming that combining both yields the best results. The method is not overly sensitive to the exact ratio of local to global advantages, with the **1:1 combination** providing the best trade-off.
* **Local + Global (1:1)**: **65.40** average score, significantly improving performance compared to using only local or global advantages.
### 4. Robustness to Misleading Contexts
Our method demonstrates robust performance even in the presence of misleading contexts, with strong improvements over GRPO w/ RAG in the presence of conflicting passages.
* **With 1 misleading passage**: Ours 43.94% vs. GRPO w/ RAG 26.94%
* **With 8 misleading passages**: Ours 40.07% vs. GRPO w/ RAG 25.42%
### 5. Context Dependence vs. Coincidence
We explicitly split the test set into **TIFE** (where internal knowledge is correct) and **FITE** (where context is correct), demonstrating that our method effectively learns to rely on the correct knowledge source (PK or CK) rather than relying on coincidence.

**TIFE**: Ours 66.00% vs. GRPO w/ RAG 62.50%

**FITE**: Ours 73.05% vs. GRPO w/ RAG 72.04%

Additionally, we introduced two diagnostic metrics: **KL divergence** and **factuality scores**. These metrics show that our method aligns much less with misleading context and stays closer to **parametric knowledge (PK)** when the context is wrong.

**Factuality-score**: Ours 0.16 vs. GRPO w/ RAG 0.40

**KL-score**: Ours 168.83 vs. GRPO w/ RAG 213.73
### 6. Query-only vs. Query+Context Behavior
**Our method benefits from context when it is correct and mitigates it when it is wrong.**
* When context is correct (S1), using context improves our method from 35.64% (query-only) to 75.08%, well above all query-only baselines, showing that the policy indeed exploits high-quality context.
* When context is wrong (S2), using context still improves our method from 36.70% to 43.94%, and substantially outperforms GRPO-RAG with context (26.94%), showing that the policy has learned to attenuate misleading context rather than blindly relying on it.
### 7. Sensitivity to Good/Bad Context Ratios
We ran a sensitivity study varying the ratio of good vs. bad context in training, showing that our method is effective even with **99% good context**. As the ratio becomes more realistic (e.g., 1:1 or 3:1), robustness to bad context improves significantly.

* **99% Good Context**: Ours 72.11% (S1), 17.34% (S2) — improvement over GRPO w/ RAG.
* **1:1 Good/Bad Ratio**: Ours 75.08% (S1), 43.94% (S2) — large gain in S2 over GRPO w/ RAG.

Thank you again for the additional effort and for helping uphold the high standards that make ICLR such a strong and principled community.

Sincerely,

The Authors

---

### Meta-Review · Area_Chair_UN3v · 2026-01-08

**Summary:**

This paper addresses context interference in retrieval-augmented generation, a practical problem where LLMs over-rely on noisy or misleading retrieved passages. The proposed Knowledgeable-R1 framework introduces an RL approach with joint PK/CK/RPK sampling.

Reviewers were happy with the problem motivation, comprehensive ablations, and strong empirical gains on counterfactual and conflicting contexts. The rebuttal addressed key concerns.

For the camera-ready version, authors should apply promised revisions, including:
- clean up notations
- discuss the S1 gap and the weighting trade-off
- move retrieval setup details from the appendix to the main
- include the $\beta$-sensitivity study and training dynamics
- fix other presentation issues

**Reviewer Concerns:**

1. Scope and realism [GFpR, 6P9g, C4dV]

[Reviewers] The method appears to rely heavily on samples with "bad context" in training. If 99% of training data is good context, can the method still learn to resist erroneous context? [6P9g] Limited exploration of partial-conflict granularity; sensitivity analysis on varying proportions of misleading passages is missing [C4dV]. Evaluation focuses on QA-style tasks; unclear whether robustness generalises to open-ended generation, dialogue, or summarisation [C4dV]. The inference template always feeds query+context; comparison to query-only inference is missing [GFpR].

[Authors] Provide a sensitivity study on good/bad context ratios. Even with 99% good context, the method still improves S2 accuracy (17.34% vs 14.31% for GRPO). With 1:1 ratio, gains are substantial (43.94% vs 26.94%). Acknowledge QA focus; broader tasks left for future work.

[AC] Reasonable response.


2. $\beta$ scheduling and sensitivity analysis [GFpR, 72rj, C4dV]

[Reviewers] The paper reports that fixed $\beta$ degrades performance but provides no sensitivity analysis [GFpR]. Asked to compare fixed $\beta$ settings (0.2, 0.5, 0.8, 1.0) versus adaptive $\beta$ to demonstrate the necessity of adaptivity [72rj]. The adaptive penalty scaling is heuristic with limited theoretical justification [C4dV].

[Authors] Provided a sensitivity study. Different datasets have different optimal fixed $\beta$ values, while adaptive $\beta$ achieves the best or near-best on all three. Also have shown $\beta$ dynamics during training: $\beta$ quickly converges to a low range (e.g., 0.23 → 0.01 over 8 steps) that strongly attenuates negative advantages for RPK samples when context is erroneous.

[AC] The empirical study seems to address the reviewers' requests.

3. Method presentation [6P9g, C4dV]

[Reviewers] The definition of RPK sampling is confusing. The notation suggests conditioning on query+context while also referencing query-only outputs. The same variable seems to denote different things [C4dV]. It is unclear whether PK, CK, and RPK are three separate controllers or training scaffolds for a single model. If only one model exists at inference, how does it arbitrate between conflicting behaviours? [6P9g]

[Authors] Clarified the details.

[AC] The authors' clarification is reasonable. Suggest to thin notations.

4. Experiments [GFpR, 72rj, 6P9g]

[Reviewers] Retrieval details are unclear: which retriever, indexing corpus, and query formulations were used? [GFpR] The "23% over GRPO" claim in the abstract is ambiguous; aggregation method unclear [GFpR]. The paper relies on exact match accuracy but does not directly quantify context dependence [GFpR]. Include finetuning methods cited in Related Work as baselines [72rj]. Add a stronger SFT baseline trained on the same mixed-context data with explicit instructions [6P9g].

[Authors] Clarified: oracle contexts are used without retrieval. Authors will revise wording to avoid confusion with relative gains. Added context-dependence diagnostics. Self-RAG and InFO-RAG require expert reasoning chains absent from these datasets; instead provide SFT baselines on the same mixed-context data with explicit instructions - SFT alone seems insufficient.

[AC] The clarifications on setup and evaluation seem reasonable.

**Reviewer Scores:**

GFpR: 6 > 6
All concerns addressed.

72rj: 4 > 6
Two main concerns:
- S1 performance gap
- Advantage composition ablation

seem to have been addressed. S1 gap is tunable via component weighting (1:2:1 closes the gap). Combining local+global advantages helps (+4.45 pp over local-only).

6P9g: 6 > 6
Concerns addressed.

C4dV: 6 > 6
Concerns addressed.

---

### Decision · Program_Chairs · 2026-01-26

Accept (Poster)